# Mapping transiently formed and sparsely populated conformations on a complex energy landscape

Yong Wang, Elena Papaleo, Kresten Lindorff-Larsen*

Structural Biology and NMR Laboratory, Linderstrøm-Lang Centre for Protein Science, Department of Biology, University of Copenhagen, Copenhagen, Denmark

**Abstract** Determining the structures, kinetics, thermodynamics and mechanisms that underlie conformational exchange processes in proteins remains extremely difficult. Only in favourable cases is it possible to provide atomic-level descriptions of sparsely populated and transiently formed alternative conformations. Here we benchmark the ability of enhanced-sampling molecular dynamics simulations to determine the free energy landscape of the L99A cavity mutant of T4 lysozyme. We find that the simulations capture key properties previously measured by NMR relaxation dispersion methods including the structure of a minor conformation, the kinetics and thermodynamics of conformational exchange, and the effect of mutations. We discover a new tunnel that involves the transient exposure towards the solvent of an internal cavity, and show it to be relevant for ligand escape. Together, our results provide a comprehensive view of the structural landscape of a protein, and point forward to studies of conformational exchange in systems that are less characterized experimentally.

*For correspondence: lindorff@
bio.ku.dk

## Introduction

Proteins are dynamical entities whose ability to change shape often plays essential roles in their function. From an experimental point of view, intra-basin dynamics is often described via conformational ensembles whereas larger scale (and often slower) motions are characterized as a conformational exchange between distinct conformational states. The latter are often simplified as a two-site exchange process, $G{\rightleftharpoons}E$, between a highly populated ground (G) state, and a transiently populated minor (or 'excited', E) state. While the structure of the ground state may often be determined by conventional structural biology tools, it is very difficult to obtain atomic-level insight into minor conformations due to their transient nature and low populations. As these minor conformations may, however, be critical to protein functions, including protein folding, ligand binding, enzyme catalysis, and signal transduction (*Mulder et al., 2001*; *Tang et al., 2007*; *Baldwin and Kay, 2009*) it is important to be able to characterize them in detail. While it may in certain cases be possible to capture sparsely populated conformations in crystals under perturbed experimental conditions, or to examine their structures by analysis of electron density maps (*Fraser et al., 2009*), NMR spectroscopy provides unique opportunities to study the dynamical equilibrium between major and minor conformations (*Baldwin and Kay, 2009*; *Sekhar and Kay, 2013*) via e.g. chemical-exchange saturation transfer (*Vallurupalli et al., 2012*), Carr-Purcell-Meiboom-Gill (CPMG) relaxation dispersion (*Hansen et al., 2008*), or indirectly via paramagnetic relaxation enhancement (*Tang et al., 2007*) or residual dipolar coupling (*Lukin et al., 2003*) experiments. In favourable cases such experiments can provide not only thermodynamic and kinetic information (i.e. the population of G and E states and the rate of exchange between them), but also structural information in the form of

**eLife digest** Proteins are the workhorses of cells, where they perform a wide range of roles. To do so, they adopt specific three-dimensional structures that enable them to interact with other molecules as necessary. Often a protein needs to be able to shift between different states with distinct structures as it goes about its job. To fully understand how a protein works, it is important to be able to characterize these different structures and how the protein changes between them.

Many of the experimental techniques used to study protein structure rely on isolating the individual structural forms of a protein. Since many structures only exist briefly, this can be very difficult. To complement experimental results, computer simulations allow researchers to model how atoms behave within a molecule. However, a number of factors limit how well these models represent what happens experimentally, such as the accuracy of the physical description used for the modeling.

Wang et al. set out to test and benchmark how well computer simulations could model changes in structure for a protein called T4 lysozyme, which has been studied extensively using experimental techniques. T4 lysozyme exists in two different states that have distinct structures. By comparing existing detailed experimental measurements with the results of their simulations, Wang et al. found that the simulations could capture key aspects of how T4 lysozyme changes its shape.

The simulations described the structure of the protein in both states and accurately determined the relative proportion of molecules that are found in each state. They could also determine how long it takes for a molecule to change its shape from one state to the other. The findings allowed Wang et al. to describe in fine detail – down to the level of individual atoms – how the protein changes its shape and how mutations in the protein affect its ability to do so. A key question for future studies is whether these insights can be extended to other proteins that are less well characterized experimentally than T4 lysozyme.

chemical shifts (CS), that can be used to determine the structure of the transiently populated state (*Sekhar and Kay, 2013*).

Despite the important developments in NMR described above, it remains very difficult to obtain structural models of minor conformations, and a substantial amount of experiments are required. Further, it is generally not possible to use such experiments to infer the mechanisms of interconversion, and to provide a more global description of the multi-state free energy landscape (*Zhuravlev and Papoian, 2010*; *Wang et al., 2012*). In the language of energy landscape theory (*Onuchic et al., 1997*), free energy basins and their depths control the population and stability of functionally distinct states, while the relative positions of basins and the inter-basin barrier heights determine the kinetics and mechanism of conformational exchange. As a complement to experiments, such functional landscapes can be explored by in silico techniques, such as molecular dynamics (MD) simulations, that may both be used to help interpret experimental data and provide new hypotheses for testing (*Karplus and Lavery, 2014*; *Eaton and Muñoz, 2014*). Nevertheless, the general applicability of simulation methods may be limited by both the accuracy of the physical models (i.e. force fields) used to describe the free energy landscape and our ability to sample these efficiently by computation. We therefore set out to benchmark the ability of simulations to determine conformational free energy landscapes.

The L99A variant of lysozyme from the T4 bacteriophage (T4L) has proven an excellent model system to understand protein structure and dynamics. Originally designed a 'cavity creating' variant to probe protein stability (*Eriksson et al., 1992b*) it was also demonstrated that the large (150 Å$^3$) internal cavity can bind hydrophobic ligands such as benzene (*Eriksson et al., 1992a*; *Liu et al., 2009*). It was early established that the cavity is inaccessible to solvent in the ground state, but that ligand binding is rapid (*Feher et al., 1996*), suggesting protein dynamics to play a potential role in the binding process. This posts a long-standing question of how the ligands gain access to the buried cavity (*Mulder et al., 2000*; *López et al., 2013*; *Merski et al., 2015*; *Miao et al., 2015*).

NMR relaxation dispersion measurements of L99A T4L demonstrated that this variant, but not the wild type protein, displayed conformational exchange on the millisecond timescale between the

ground state and a minor state populated at around 3% (at room temperature) (**Mulder et al., 2001**). Such small populations generally lead only to minimal perturbations of ensemble-averaged experimental quantities making structural studies difficult, and hence it was difficult to probe whether the exchange process indeed allowed for ligand access to the cavity. A series of additional relaxation dispersion experiments, however, made it possible to obtain backbone and side chain CSs of the minor E state of L99A (**Mulder et al., 2002**; **Bouvignies et al., 2011**). The backbone CS data were subsequently used as input to a CS-based structure refinement protocol (CS-ROSETTA) to produce a structural model of the E state ($E_{ROSETTA}$; **Figure 1**) of the L99A mutant (**Bouvignies et al., 2011**). This model was based in part on the crystal structure of the ground state of L99A (referred to in what follows as $G_{Xray}$), but perturbing the structure in regions that the experiments demonstrated to undergo conformational change in a way so that the final model ($E_{ROSETTA}$) agrees with experiments. The structure was further validated by creating and solving the structure of a triple mutant variant that inverts the populations of the G and E states. The $E_{ROSETTA}$ structure revealed substantial local rearrangements in T4L L99A, in particular near the cavity which gets filled by the side chain of a phenylalanine at position 114 ($F_{114}$). Because the cavity is filled and solvent inaccessible in the E-state, the structure did, however, not reveal how ligands might access the cavity.

In an attempt to benchmark the ability of simulations to map conformational free energy landscapes, we have here employed a series of in silico experiments designed to probe the structure and dynamics of L99A T4L and have compared the results to NMR measurements. We used enhanced-sampling MD simulations in explicit solvent and with state-of-the-art force fields to map the free-energy landscape including the exchange between the major and minor conformations of the protein. We used a series of recently developed metadynamics methods (**Laio and Parrinello, 2002**) to sample the conformational exchange process and associated structure and thermodynamics, as well as to determine the kinetics and mechanisms of exchange. We obtained additional insight into the structural dynamics of the E state using simulations that employed the experimental CSs as replica-averaged restraints. Our results provide a coherent picture of the conformational dynamics in L99A and extend the insights obtained from recent simulations of a triple mutant of T4L (**Vallurupalli et al., 2016**), by providing new information about the mechanisms of exchange and the transient exposure of the internal cavity. Together with previous results for Cyclophilin A (**Papaleo et al., 2014**) the results described here reiterate how simulation methods have now reached a stage where they can be used to study slow, conformational exchange processes such as those probed by NMR relaxation dispersion even in cases where less information is available from experiments.

## Results and discussion

### Mapping the free-energy landscape

As the average lifetime of the G and E states are on the order of 20–50 ms and 1 ms, respectively (**Mulder et al., 2001**, **2002**; **Bouvignies et al., 2011**), direct and reversible sampling of the G-E transition at equilibrium would be extremely demanding computationally. Indeed, a recent set of simulations of a triple mutant of T4L, which has a substantially faster kinetics, was able only to observe spontaneous transitions in one direction (**Vallurupalli et al., 2016**). We therefore resorted to a set of flexible and efficient enhanced sampling methods, collectively known as 'metadynamics' (**Laio and Parrinello, 2002**), that have previously been used in a wide range of applications. In metadynamics simulations, a time-dependent bias is continuously added to the energy surface along a small number of user-defined collective variables (CVs). In this way, sampling is enhanced to reach new regions of conformational space and at the same time allows one to reconstruct the (Boltzmann) free-energy surface. The success of the approach hinges on the ability to find a set of CVs that together describe the slowly varying degrees of freedom and map the important regions of the conformational landscape.

We first performed a set of metadynamics simulations in the well-tempered ensemble (**Barducci et al., 2008**) using so-called path CVs ($S_{path}$ and $Z_{path}$) (**Branduardi et al., 2007**; **Saladino et al., 2012**) with the aid of recently developed adaptive hills to aid in the convergence of the sampling (**Branduardi et al., 2012**; **Dama et al., 2014**) (see details in Appendix and **Appendix 1—table 1**). In short, the $S_{path}$ variable describes the progress of the conformational transition

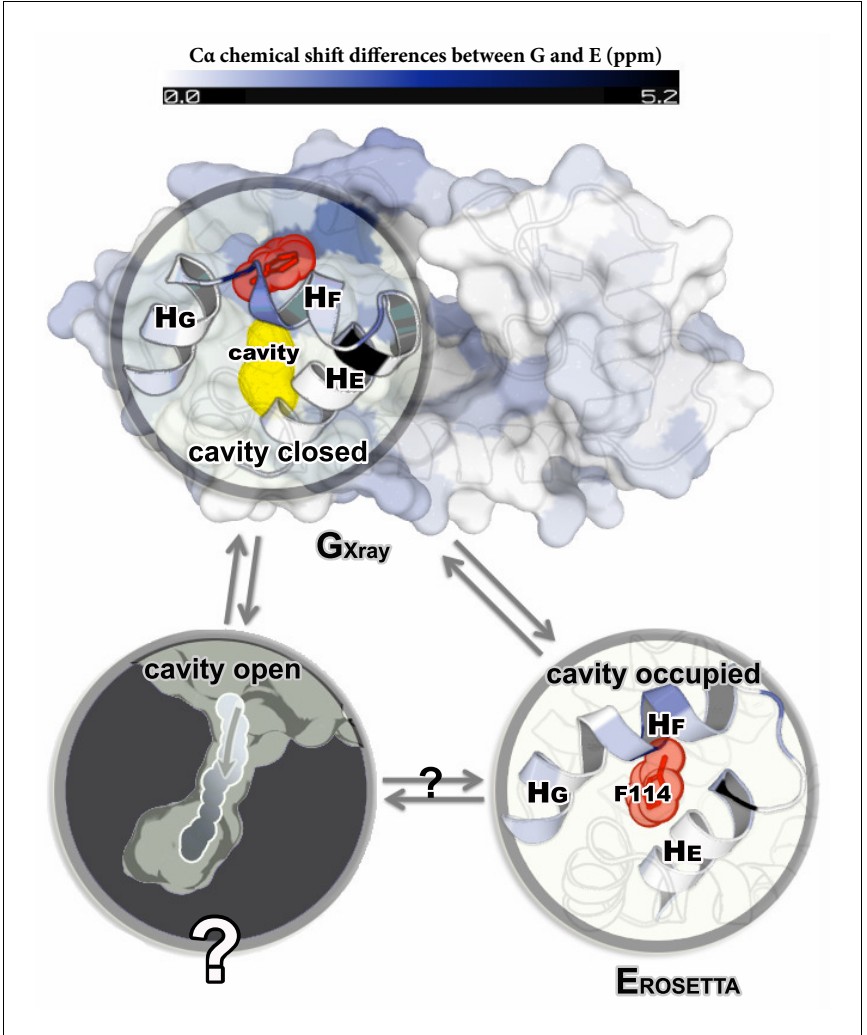

**Figure 1.** Structures of the major G and minor E states of L99A T4L and the hidden state hypothesis. The X-ray structure of the G state ($G_{Xray}$; PDB ID code 3DMV) has a large internal cavity within the core of the C-terminal domain that is able to bind hydrophobic ligands. The structure of the E state ($E_{ROSETTA}$; PDB ID code 2LC9) was previously determined by CS-ROSETTA using chemical shifts. The G and E states are overall similar, apart from the region surrounding the internal cavity. Comparison of the two structures revealed two remarkable conformational changes from G to E: helix F (denoted as $H_F$) rotates and fuses with helix G ($H_G$) into a longer helix, and the side chain of phenylalanine at position 114 ($F_{114}$) rotates so as to occupy part of the cavity. As the cavity is inaccessible in both the $G_{Xray}$ and $E_{ROSETTA}$ structures it has been hypothesized that ligand entry occurs via a third 'cavity open' state (*Merski et al., 2015*).

between the $G_{Xray}$ and $E_{ROSETTA}$ structures with additional 'interpolation' using an optimal 'reference' path in a simplified model (see details in Appendix and *Figure 2—figure supplement 1*), while $Z_{path}$ measures the distance to this reference path. In this way, the two-dimensional free energy landscape along $S_{path}$ and $Z_{path}$ provides a useful description on conformational exchange between ground and excited states that does not assume that the initial reference path describes perfectly the actual path(s) taken.

Projecting the sampled free energy landscape along $S_{path}$ (upper panel of *Figure 2*) reveals a deep, narrow free energy basin around $S_{path} = 0.2$ (labeled by red sphere and corresponding to the G state), and a broader, shallow free energy basin with $S_{path}$ ranging from 0.6 to 0.8 (labeled by blue sphere and corresponding to the E state). Additional information is obtained from the two-dimensional landscape (shown as a negative free energy landscape, -F($S_{path}$, $Z_{path}$), in the lower panel of

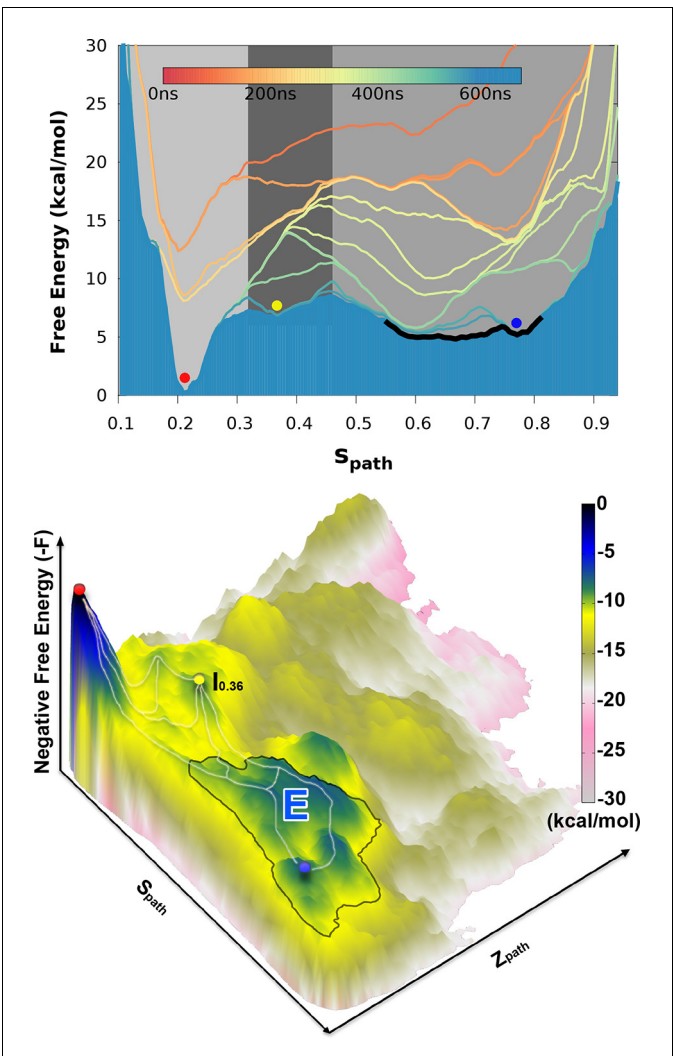

**Figure 2.** Free energy landscape of the L99A variant of T4L. In the upper panel, we show the projection of the free energy along $S_{path}$, representing the Boltzmann distribution of the force field employed along the reference path. Differently colored lines represent the free energy profiles obtained at different stages of the simulation, whose total length was 667ns. As the simulation progressed, we rapidly found two distinct free energy basins, and the free energy profile was essentially constant during the last 100 ns of the simulation. Free energy basins around $S_{path} = 0.2$ and $S_{path} = 0.75$ correspond to the previously determined structures of the G- and E-state, respectively (labelled by red and blue dots, respectively). As discussed further below, the E-state is relatively broad and is here indicated by the thick, dark line with $S_{path}$ ranging from 0.55 to 0.83. In the lower panel, we show the three-dimensional negative free energy landscape, $-F(S_{path}, Z_{path})$, that reveals a more complex and rough landscape with multiple free energy minima, corresponding to mountains in the negative free energy landscape. An intermediate-state basin around $S_{path} = 0.36$ and $Z_{path} = 0.05$ nm$^2$, which we denote $I_{0.36}$, is labeled by a yellow dot.

The following figure supplements are available for figure 2:

**Figure supplement 1.** Approximately equidistant frames along the reference path.

**Figure supplement 2.** One and two dimensional free energy landscape of L99A and the triple mutant.

*Figure 2*) which reveals a complex and rough landscape with multiple free energy minima

(corresponding to mountains in the negative free energy landscape). Subsequently, structural inspection of these minima identified that the conformations in the basins around $S_{path} = 0.2$ and $S_{path} = 0.75$ correspond to the structures of $G_{Xray}$ and $E_{ROSETTA}$, respectively.

The broad nature of the free energy landscape in the region of the minor state is consistent with the observation that our MD simulations initiated from $E_{ROSETTA}$ display significant conformational fluctuations (RUN20 and RUN22 in *Appendix 1—table 1*). Furthermore, our metadynamics simulations revealed multiple local free energy minima adjacent to the $E_{ROSETTA}$ basin, together composing a wider basin (highlighted by the black curve in *Figure 2*). Thus, these simulations suggest that the E state displays substantial conformational dynamics, a result corroborated by simulations that have been biased by the experimental data (see section 'Simulations of the minor state using chemical shift restraints').

In addition to free-energy minima corresponding to the $G$ and $E$ states, we also found a free energy minimum around $S_{path} = 0.36$ and $Z_{path} = 0.05$ nm$^2$ (denoted as $I_{0.36}$ and labeled by a yellow sphere in *Figure 2*) that is located between the G and E states on the one-dimensional free-energy surface. We note, however, that it is difficult to infer dominant reaction pathways from such free energy surfaces, and so from this data alone, we cannot determine whether $I_{0.36}$ occurs as an intermediate in G-E conformational transitions. Indeed, it appears from the two-dimensional surface that there exist multiple possible pathways between G and E, as illustrated by white lines along the mountain ridges of the negative free energy landscape in the lower panel of *Figure 2*. (We also explored the mechanism of exchange by reconnaissance metadynamics simulations (*Tribello et al., 2011*), the results of which are described and discussed further below.)

## Effect of mutations on the free energy landscape

Based on the encouraging results above for L99A T4L, we examined whether simulations could also capture the effect of mutations on the free energy landscape. Using Rosetta energy calculations on the $G_{Xray}$ and $E_{ROSETTA}$ structures it was previously demonstrated that two additional mutations, G113A and R119P, when introduced into the L99A background, cause an inversion in the populations of the two states (*Bouvignies et al., 2011*; *Vallurupalli et al., 2016*). Indeed, NMR data demonstrated that the triple mutant roughly inverts the populations of the two states so that the minor state structure (of L99A) now dominates (with a 96% population) the triple mutant. We repeated the calculations described above for L99A also for the triple mutant. Remarkably, the free energy profile of the triple mutant obtained using metadynamics simulations reveals a free energy landscape with a dominant minimum around $S_{path}$=0.7 and a higher energy conformation around $S_{path}$=0.15 (*Figure 2—figure supplement 2*). Thus, like our previous observations for a 'state-inverting mutation' in Cyclophilin A (*Papaleo et al., 2014*), we find here that the force field and the sampling method are sufficiently accurate to capture the effect of point mutations on the free energy landscape. Further, we note that the barrier height for the conformational exchange in the triple mutant is very similar to the value recently estimated using a completely orthogonal approach (*Vallurupalli et al., 2016*). Finally, we attempted to determine the free energy landscape of the L99A,G113A double mutant, which has roughly equal populations of the two states (*Bouvignies et al., 2011*), but this simulation did not converge on the simulation timescales at which the two other variants converged.

## Calculating conformational free energies

With a free-energy surface in hand and a method to distinguish G- and E-state conformations, we calculated the free energy difference, $\Delta G$, between the two conformational states, and compared with the experimental values. We divided the global conformational space into two coarse-grained states by defining the separatrix at $S_{path} = 0.46$ which corresponds to a saddle point on the free energy surface, on the basis of the observations above that the E state is relatively broad. Although a stricter definition of how to divide the reaction coordinate certainly helps the precise calculation, here we just used this simple definition to make an approximate estimation of the free energy difference. Further, since the barrier region is sparsely populated, the exact point of division has only a modest effect on the results. By summing the populations on the two sides of the barrier we calculated $\Delta G$ as a function of the simulation time (*Figure 3*). Initially during the simulations the free energy profile varies substantially (*Figure 2*) and the free energy difference equally fluctuates. As the simulations converge, however, the free energy difference between the two states stabilize to a

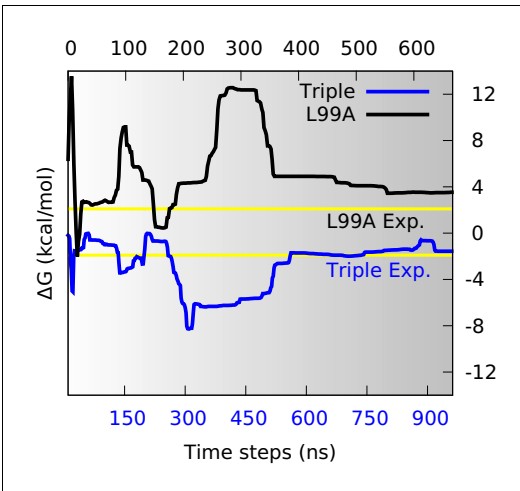

**Figure 3.** Estimation of free energy differences and comparison with experimental measurements. We divided the global conformational space into two coarse-grained states by defining the separatrix at $S_{path} = 0.46$ (0.48 for the triple mutant) in the free energy profile (*Figure 2—figure supplement 2*) which corresponds to a saddle point of the free energy surface, and then estimated the free energy differences between the two states ($\Delta G$) from their populations. The time evolution of $\Delta G$ of L99A (upper time axis) and the triple mutant (lower axis) are shown as black and blue curves, respectively. The experimentally determined values (2.1 kcal mol$^{-1}$ for L99A and $-1.9$ kcal mol$^{-1}$ for the triple mutant) are shown as yellow lines.

value at approximately $\Delta G$=3.5 kcal mol$^{-1}$ (*Figure 3*, black line). This value can be compared to the value of 2.1 kcal mol$^{-1}$ obtained from NMR relaxation dispersion experiments (*Mulder et al., 2001*), revealing reasonably good, albeit not exact, agreement with the experiments.

Similar calculations using the simulations of the triple mutant also converge, in this case to about $-1.6$ kcal mol$^{-1}$ (*Figure 3*, blue line), in excellent agreement with the experimental measurement ($-1.9$ kcal mol$^{-1}$) (*Bouvignies et al., 2011*). Combining these two free energy differences we find that the G113A, R119P mutations cause a shift in the G-E free energy of 5.1 kcal mol$^{-1}$ in simulations compared to 4.0 kcal mol$^{-1}$ obtained by experiments. Thus, we find that the simulations with reasonably high accuracy are able to capture the thermodynamics of the conformational exchange between the two states. While the generality of such observations will need to be established by additional studies, we note here that comparably good agreement was obtained when estimating the effect of the S99T mutations in Cyclophilin A (*Papaleo et al., 2014*).

In our previous work on Cyclophilin A (*Papaleo et al., 2014*) we sampled the conformational exchange using parallel-tempering metadynamics simulations (*Bonomi and Parrinello, 2010*) using four CVs that we chose to describe the structural differences between the G and E states in that protein. We note here that we also tried a similar approach here but unfortunately failed to observe a complete G-to-E transition, even in a relatively long trajectory of about 1 μs per replica (CVs summarized in *Appendix 1—table 2*, parameters shown in *Appendix 1—table 1*). This negative result is likely due to the CVs chosen did not fully capture the relevant, slowly changing degrees of freedom, thus giving rise to insufficient sampling even with the use of a parallel tempering scheme.

## Calculating the rates of conformational exchange

Enhanced-sampling simulations such as those described above provide an effective means of mapping the free-energy landscape and hence the structural and thermodynamic aspects of conformational exchange. While the same free-energy landscape also determines the kinetics and mechanisms of exchange it may be more difficult to extract this information from e.g. path-CV-based metadynamics (PathMetaD) simulations. To examine how well simulations can also be used to determine the rates of the G-to-E transitions, quantities that can also be measured by NMR, we used the recently developed 'infrequent metadynamics' method (InMetaD, see details in Appendix) (*Tiwary and Parrinello, 2013*; *Salvalaglio et al., 2014*; *Tiwary et al., 2015a*; *2015b*). Briefly described, the approach calculates first-passage times for the conformational change in the presence of a slowly-added bias along a few CVs, here chosen as the path CVs also used to map the landscape. By adding the bias slowly (and with lower amplitude) we aim to avoid biasing the transition-state region and hence to increase the rate only by lowering the barrier height; in this way it is possible to correct the first-passage times for the bias introduced.

Using this approach on L99A T4L we collected 42 and 36 independent trajectories with state-to-state transition starting from either the $G$ state or $E$ state, respectively (*Appendix 1—figure 1* and *Appendix 1—figure 2* ). The (unbiased) rates that we calculated (*Table 1* and *Appendix 1—figure*

3) are in good agreement with the experimental rates (*Mulder et al., 2001*; *Bouvignies et al., 2011*) (within a factor of 10), corresponding to an average error of the barrier height of ~1 kcal mol$^{-1}$. We also performed similar calculations for the 'population-inverting' triple mutant, where we collected 30 transitions (15 for each direction) using InMetaD simulations. As for L99A, we also here find similarly good agreement with experimental measurements (*Vallurupalli et al., 2016*) (*Table 1* and *Appendix 1—figure 4*). We estimated the reliability of this computational approach using a Kolmogorov-Smirnov test to examine whether the first-passage times conform to the expected Poisson distribution (*Salvalaglio et al., 2014*), and indeed the results of this analysis suggest good agreement (*Table 1*, *Appendix 1—figure 5* and *Appendix 1—figure 6*).

The ability to calculate forward and backward rates between $G$ and $E$ provided us with an alternative and independent means to estimate the free energy difference between the two states (*Table 1*), and to test the two-state assumption used in the analysis of the experimental NMR data. We therefore calculated the free energy difference from the ratio of the forward and backward reaction rates. The values obtained (2.9±0.5 kcal mol$^{-1}$ and −1.2±1.1 kcal mol$^{-1}$ for L99A and the triple mutant, respectively) are close both to the values obtained above from the equilibrium free energy landscape (3.5 kcal mol$^{-1}$ and −1.6 kcal mol$^{-1}$) and experiment (2.1 kcal mol$^{-1}$ and −1.9 kcal mol$^{-1}$). In particular, the relatively close agreement between the two independent computational estimates lends credibility both to the free energy landscape and the approach used to estimate the kinetics. The observation that both values for L99A are slightly larger than the experimental number suggests that this discrepancy (ca. 1 kcal mol$^{-1}$) can likely be explained by remaining force field deficiencies rather than lack of convergence or the computational approach used.

## Simulations of the minor state using chemical shift restraints

While the simulations described above used available structural information of G and E states to guide and enhance conformational sampling, the resulting free energy surfaces represent the Boltzmann distributions of the force field and are not otherwise biased by experimental data. To further refine the structural model of the E state we used the relaxation-dispersion derived CSs that were used to determine of $E_{ROSETTA}$ [BMRB (*Ulrich et al., 2008*) entry 17604] as input to restrained MD simulations. In these simulations, we used the experimental data as a system-specific force field correction to derive an ensemble of conformations that is compatible both with the force field and the CSs. Such replica-averaged simulations use the experimental data in a minimally-biased way that is consistent with the Principle of Maximum Entropy (*Pitera and Chodera, 2012*; *Roux and Weare, 2013*; *Cavalli et al., 2013*; *Boomsma et al., 2014*).

We performed CS-restrained MD simulations of the $E$ state of L99A averaging the CSs over four replicas. Although the number of replicas is a free parameter, which should in principle be chosen as large as possible, it has been demonstrated that four replicas are sufficient to reproduce the structural heterogeneity accurately (*Camilloni et al., 2013*) without excessive computational requirements. The agreement between calculated and experimental CSs was quantified by the root-mean-square deviation between the two (*Figure 4—figure supplement 1*). In particular, it is important not to bias the agreement beyond what can be expected based on the inherent accuracy of the CS

**Table 1.** Free energy differences and rates of conformational exchange.

| | $\tau_{G \to E}$ (ms) | $\tau_{E \to G}$ (ms) | $\Delta G$ (kcal mol$^{-1}$) |
|---|---|---|---|
| | L99A | | |
| NMR | 20 | 0.7 | 2.1 |
| InMetaD | 175±56 | 1.4±0.6 | 2.9±0.5 |
| PathMetaD | | | 3.5 |
| | L99A/G113A/R119P | | |
| NMR | 0.2 | 4 | -1.9 |
| InMetaD | 2.0±1.7 | 14.3±8.3 | -1.2±1.1 |
| PathMetaD | | | -1.6 |

prediction methods (we assumed that the error in the experimental CS measurement, even for the $E$ state, is negligible in comparison). Thus, we compared the experimental CS values of the minor state with the values calculated using the $E_{ROSETTA}$ structure as input to CamShift (*Kohlhoff et al., 2009*), Sparta+ (*Shen and Bax, 2010*) and ShiftX (*Neal et al., 2003*) (*Figure 4—figure supplement 2*). The average RMSDs for five measured nuclei ($H_\alpha$, $H_N$, $N$, $C'$ and $C_\alpha$) are 0.2, 0.4, 2.0, 0.8 and 1.1ppm, respectively (*Appendix 1—table 1*), which are close to the inherent uncertainty of the CS back-calculation, indicating that the level of agreement enforced is reasonable.

To compare the results of these experimentally-biased simulations with the experimentally-unbiased simulations described above, we projected the CS-restrained MD trajectories onto either one (*Figure 4*) or both (*Figure 4—figure supplement 3*) of the $S_{path}$ and $Z_{path}$ variables used in the path-variable-driven simulations (PathMetaD). The distribution of conformations obtained using the E-state CSs as restraints is in good agreement with the broad free energy profile of the E-state obtained in the metadynamics simulations that did not include any experimental restraints. To ensure that this observation is not merely an artifact of both simulations using the same force field (CHARMM22*), we repeated the biased simulations using the Amber ff99SB*-ILDN force field and obtained comparable results. We also verified that the conclusions obtained are reasonably robust to other variables such as the number of replicas and the strength of restraints (*Figure 4—figure supplement 4*).

As a final and independent test of the structural ensemble of the minor conformation of L99A we used the ground state CSs of the triple mutant (BMRB entry 17603), which corresponds structurally to the E state of L99A, as restraints in replica-averaged CS-biased simulations (*Figure 4—figure supplement 5*). Although not fully converged, these simulations also cover roughly the same region of conformational space when projected along $S_{path}$ (*Figure 4*).

Thus, together our different simulations, which employ different force fields, are either unbiased or biased by experimental data, and use either dispersion-derived (L99A) or directly obtained (triple mutant) CS all provide a consistent view of the minor E-state conformation of L99A. We also note that the CS-derived ensembles of the E-state support the way we divided the G- and E-states when calculating conformational free energy differences between the two states.

## Mechanisms of conformational exchange

Having validated that our simulations can provide a relatively accurate description of the structure, thermodynamics and kinetics of conformational exchange we proceeded to explore the molecular mechanism of the G-to-E transitions. We used the recently developed reconnaissance metadynamics approach (*Tribello et al., 2010*), that was specifically designed to enhance sampling of complicated conformational transitions and has been employed to explore the conformational dynamics of complex systems (*Tribello et al., 2011*; *Söderhjelm et al., 2012*).

We performed three independent reconnaissance metadynamics simulations of L99A starting from the G state (summarized in *Appendix 1—table 1*) using the same geometry-based CVs that we also used in the parallel-tempering simulations described above. We observed complete conformational transitions from the G to E state in the reconnaissance simulations in as little as tens of nanoseconds of simulations (*Figure 5—figure supplement 1*)— at least 1–2 orders of magnitude faster than standard metadynamics. These G-to-E and E-to-G transitions, although biased by the CVs, provide insight into the potential mechanisms of exchange. To ease comparison with the equilibrium sampling of the free energy landscape we projected these transitions onto the free energy surface $F(S_{path},Z_{path})$ (*Figure 5*). The results reveal multiple possible routes connecting the G and E states, consistent with the multiple gullies found on the free energy surface (*Figure 2*). The trajectories also suggested that the G-to-E interconversion can either take place directly without passing the $I_{0.36}$ state or indirectly via it.

In the context of coarse-grained kinetic models the results above would suggest at least two possible mechanisms operate in parallel: $G{\rightleftharpoons}E$ or $G{\rightleftharpoons}I_{0.36}{\rightleftharpoons}E$. Further inspection of the structures along these different kinetics routes (see the trajectories of other order parameters in *Figure 5—figure supplement 2* and *Videos 1–4*) suggested an interesting distinction between the two. In the $G{\rightleftharpoons}I_{0.36}{\rightleftharpoons}E$ route the side chain of F114, which occupies the cavity in the E state, gets transiently exposed to solvent during the transition, whereas in the direct $G{\rightleftharpoons}E$ transitions F114 can rotate its

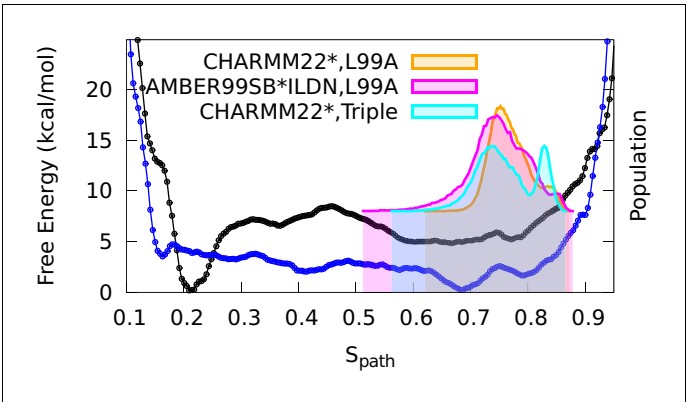

**Figure 4.** Conformational ensemble of the minor state as determined by CS biased, replica-averaged simulations. We determined an ensemble of conformations corresponding to the E-state of L99A T4L using replica-averaged CSs as a bias term in our simulations. The distribution of conformations was projected onto the $S_{path}$ variable (orange) and is compared to the free energy profile obtained above from the metadynamics simulations without experimental biases (black line). To ensure that the similar distribution of conformations is not an artifact of using the same force field (CHARMM22*) in both simulations, we repeated the CS-biased simulations using also the Amber ff99SB*-ILDN force field (magenta) and obtained similar results. Finally, we used the ground state CSs of a triple mutant of T4L, which was designed to sample the minor conformation (of L99A) as its major conformation, and also obtained a similar distribution along the $S_{path}$ variable (cyan).

The following figure supplements are available for figure 4:

**Figure supplement 1.** Equilibrium sampling of conformational regions of the E state of L99A by CS-restrained replica-averaged simulation.

**Figure supplement 2.** Estimation of the inherent uncertainty of the chemical shift calculation by different algorithms: CamShift (*Kohlhoff et al., 2009*), ShiftX (*Neal et al., 2003*) and Sparta+ (*Shen and Bax, 2010*).

**Figure supplement 3.** Force field dependency of the replica averaged MD simulations of L99A with chemical shift restraints.

**Figure supplement 4.** Effect of changing the force constant and number of replicas in CS-restrained simulation of L99A.

**Figure supplement 5.** Replica-averaged CS-restrained MD simulation of a T4L triple mutant (L99A/G113A/R119P).

---

side chain inside the protein core (see also the solvent accessible surface area calculation of F114 in *Figure 5—figure supplement 3*).

## A potential pathway for ligand binding and escape

As the internal cavity in L99A T4L remains buried in both the G and E states (and indeed occupied by F114 in the E state) it remains unclear how ligands access this internal cavity and how rapid binding and release is achieved. Visual inspection of our trajectories and solvent-accessible surface area analysis revealed structures with transient exposure of the internal cavity towards the solvent. The structures were mostly found in a region of conformational space that mapped onto the $I_{0.36}$ basin (*Figure 2*), and the events of that basin mostly took place between 430 ns and 447 ns (see *Video 5*). Thus, we mapped these structures to the free energy surface (*Figure 6—figure supplement 1*) and analysed them. Overall, the structure is more similar to the G- than E-state, though is more loosely packed. The similarity to the G-state is compatible with rapid binding and position of F114 in this state.

We used CAVER3 (*Chovancova et al., 2012*) (see parameters in *Appendix 1—table 4*) to analyse the structures and found multiple tunnels connecting the cavity with protein surface (*Figure 6—*

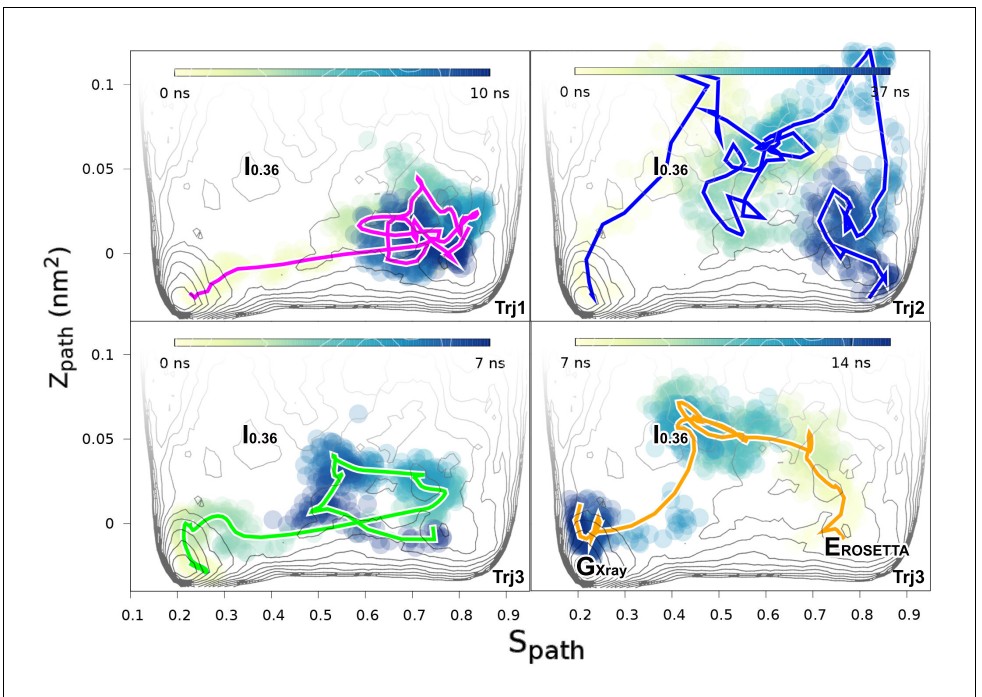

**Figure 5.** Mechanisms of the G-E conformational exchanges explored by reconnaissance metadynamics. Trajectories labeled as Trj1 (magenta), Trj2 (blue) and Trj3 (green and orange) are from the simulations RUN10, RUN11 and RUN12 (*Appendix 1—table 1*), respectively. There are multiple routes connecting the G and E states, whose interconversions can take place directly without passing the $I_{0.36}$ state or indirectly via it.

The following figure supplements are available for figure 5:

**Figure supplement 1.** Complete G-to-E transitions of L99A obtained by reconnaissance metadynamics simulations.

**Figure supplement 2.** Conformational transitions between the G and E states monitored by other order parameters.

**Figure supplement 3.** Solvent accessible surface area (SASA) calculation of the side chain of F114.

figure supplement *1* and *2*). The tunnels are relatively narrow with the typical radius of the bottle-neck (defined as the narrowest part of a given tunnel) between ~1 Å – ~2 Å. We used CAVER Analyst1.0 (*Kozlikova et al., 2014*) (see details in Appendix and parameters in *Appendix 1—table 4*) to separate the tunnels into different clusters (*Figure 6—figure supplement 3* and *Appendix 1—table 5*) with the dominant cluster (denoted tunnel#1) having a entrance located at the groove between $H_F$ and $H_I$. A typical representative structure of $I_{0.36}$ is shown in *Figure 6A*. The radii along the structures in cluster #1 vary, but share an overall shape (*Figure 6—figure supplement 1*), and we find that the maximal bottleneck radius is ~2.5 Å, the average bottleneck radius is ~1.3 Å, and the average length ~11.2 Å.

Interestingly, a series of structures of L99A were recently described, in which the internal cavity where filled with eight congeneric ligands of increasing size to eventually open the structure size (*Merski et al., 2015*). We performed a comparable tunnel analysis on those eight ligand-bound structures (PDB ID codes: 4W52 – 4W59), revealing the maximal bottleneck radius of 1.8 Å (bound with n-hexylbenzene, 4W59). Although the size of the tunnel in these X-ray structures is slightly smaller than that in $I_{0.36}$ structures, the location of the tunnel exit is consistent with the dominant tunnel#1 in $I_{0.36}$ (*Figure 6—figure supplement 3*). We note, however, that the tunnels observed in our simulation and in the ligand-induced cavity-open X-ray structure (4W59), are too narrow to allow for unhindered passage of e.g. benzene with its a van der Waals' width of 3.5 Å (*Eriksson et al.,*

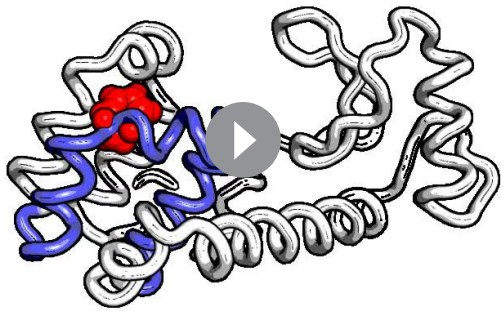 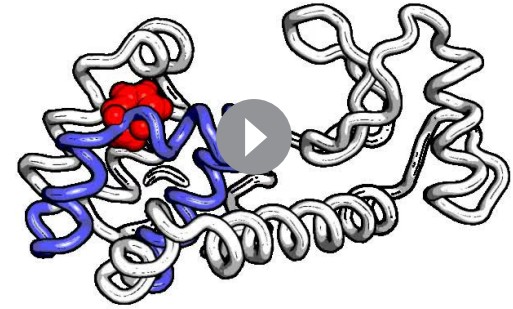

**Video 1.** Trajectory of the G-to-E conformational transition observed in Trj1, corresponding to the red trajectory in *Figure 5*. The backbone of L99A is represented by white ribbons, Helices E, F and G are highlighted in blue, while F114 is represented by red spheres.

**Video 2.** Trajectory of the G-to-E conformational transition observed in Trj2, corresponding to the blue trajectory in *Figure 5*. The backbone of L99A is represented by white ribbons, Helices E, F and G are highlighted in blue, while F114 is represented by red spheres.

*1992a*). Thus, we speculate that the transient exposure in $I_{0.36}$ might serve as a possible starting point for ligand (un)binding, which would induce (*Koshland, 1958*; *López et al., 2013*; *Wang et al., 2012*) further the opening of the tunnel.

As an initial step towards characterizing the mechanism of ligand binding and escape we used adiabatic biased molecular dynamics (ABMD) simulations (*Marchi and Ballone, 1999*; *Paci and Karplus, 1999*) to study the mechanism of how benzene escapes the internal cavity (see Appendix for details). In ABMD the system is perturbed by a 'ratcheting potential', which acts to 'select' spontaneous fluctuations towards the ligand-free state. In particular, the biasing potential is zero when the reaction coordinate (here chosen to be the RMSD of the ligand to the cavity-bound state) increases, but provides a penalty for fluctuations that brings the ligand closer to the cavity. In this way, we were able to observe multiple unbinding events in simulations despite the long lifetime (1.2 ms) of the ligand in the cavity. Most of trajectories (15 of the 20 events observed) reveal that benzene escapes from the cavity by following tunnel #1 (*Figure 6—figure supplement 4* and *Appendix 1—*

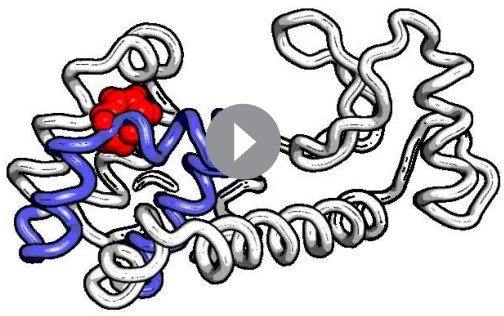 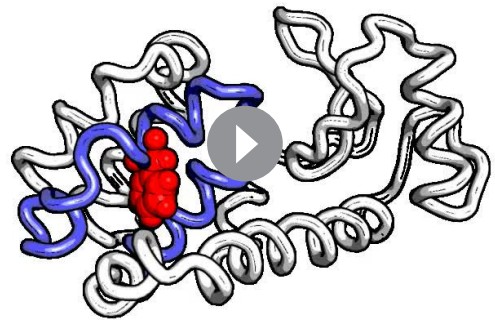

**Video 3.** Trajectory of the G-to-E conformational transition observed in Trj3, corresponding to the green trajectory in *Figure 5*. The backbone of L99A is represented by white ribbons, Helices E, F and G are highlighted in blue, while F114 is represented by red spheres.

**Video 4.** Trajectory of the E-to-G conformational transition observed in Trj3, corresponding to the yellow trajectory in *Figure 5*. The backbone of L99A is represented by white ribbons, Helices E, F and G are highlighted in blue, while F114 is represented by red spheres.

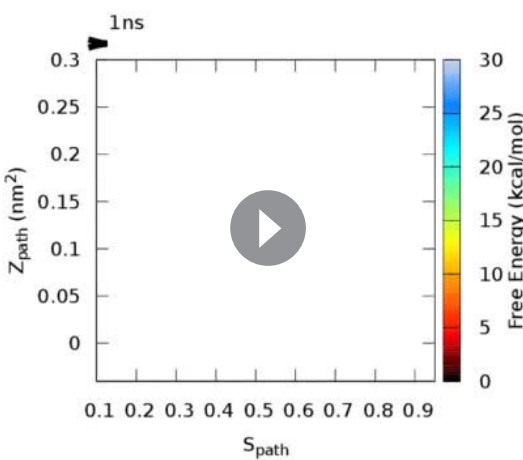

**Video 5.** Movie of the calculated two-dimensional free energy landscape of L99A as a function of simulation time. The figure shows the time evolution of the free energy surface as a function of $S_{path}$ and $Z_{path}$ sampled in a 667 ns PathMetaD simulation of L99A.

table 6). A typical unbinding path is shown in the right panel of *Figure 6* (see also *Video 6*). Because the ABMD introduces a bias to speed up ligand escape, we ensured that the observed pathway was the same at two different values of the biasing force constants (*Figure 6—figure supplement 4* and *Appendix 1—table 6*). Future work will be aimed to perform a more quantitative analysis of the ligand binding and unbinding kinetics.

## Conclusions

The ability to change shape is an essential part of the function of many proteins, but it remains difficult to characterize alternative conformations that are only transiently and sparsely populated. We have studied the L99A variant of T4L as a model system that displays a complicated set of dynamical processes which have been characterized in substantial detail. Our results show that modern simulation methods are able to provide insight into such processes, paving the way for future studies for systems that are more difficult to study experimentally.

Using a novel method for defining an initial reference path between two conformations, we were able to sample the free energy landscape described by an accurate molecular force field. In accordance with experiments, the simulations revealed two distinct free energy basins that correspond to the major and minor states found by NMR. Quantification of the free energy difference between the two states demonstrated that the force field is able to describe conformational free energies to an accuracy of about 1 kcal mol$^{-1}$. This high accuracy is corroborated by previous studies of a different protein, Cyclophilin A, where we also calculated conformational free energies and compared to relaxation dispersion experiments and found very good agreement. For both proteins we were also able to capture and quantify the effect that point mutations have on the equilibrium between the two states, and also here found good agreement with experiments. We note, however, that comparable simulations of the L99A/G113A mutant did not reach convergence.

Moving a step further, we here also calculated the kinetics of conformational exchange using a recently developed metadynamics method. For both the L99A variant and a population-inverting triple mutant we find that the calculated reaction rates are in remarkably good agreement with experiments. The ability to calculate both forward and backward rates provided us with the opportunity to obtain an independent estimate the calculated free energy difference. The finding that the free energy differences estimated in this way (for both L99A and the triple mutant) are close to those estimated from the free energy landscape provides an important validation of both approaches, and we suggest that, when possible, such calculations could be used to supplement conventional free energy estimates.

The free-energy landscape suggested that the E state is relatively broad and contains a wider range of conformations. To validate this observation, we used the same chemical shift information as was used as input to Rosetta and performed replica-averaged CS-restrained simulations. The resulting ensemble demonstrates that the experiments and force field, when used jointly, indeed are compatible with a broader E state. Thus, we suggest that the $E_{ROSETTA}$ structure and CS-restrained ensemble jointly describe the structure and dynamics of the E state.

While NMR experiments, in favourable cases, can be used to determine the structure, thermodynamics and kinetics of conformational exchange, a detailed description mechanism of interconversion remains very difficult to probe by experiments. We explored potential mechanisms of conformational exchange between the two states, finding at least two distinct routes. One route involved a direct transition with the central F114 entering the cavity within the protein, whereas a

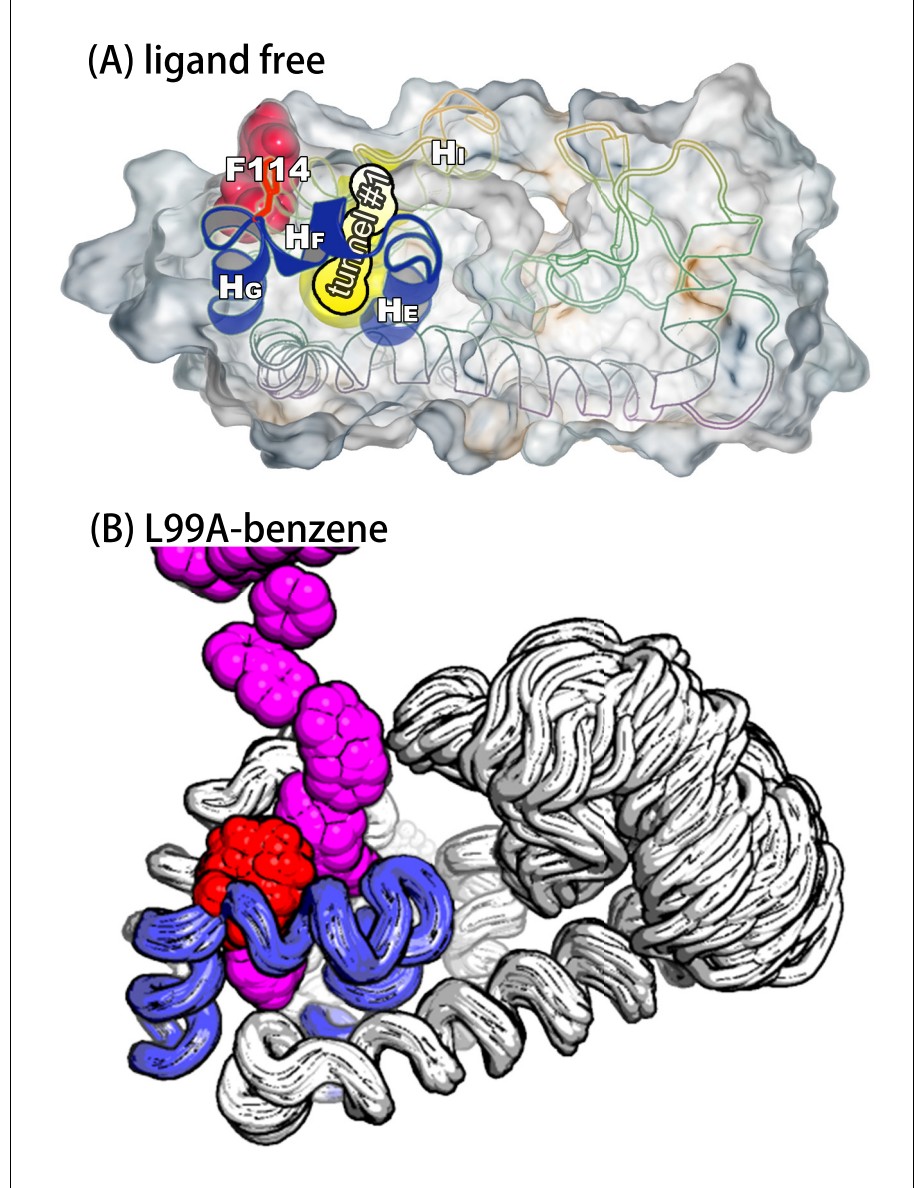

**Figure 6.** A transiently formed tunnel from the solvent to the cavity is a potential ligand binding pathway. (**A**) We here highlight the most populated tunnel structure (tunnel#1), that has an entrance located at the groove between helix F ($H_F$) and helix I ($H_I$). Helices E, F and G (blue) and F114 (red) are highlighted. (**B**) The panel shows a typical path of benzene (magenta) escaping from the cavity of L99A, as seen in ABMD simulations, via a tunnel formed in the same region as tunnel #1 (see also *Video 6*).

The following figure supplements are available for figure 6:

**Figure supplement 1.** A transiently formed tunnel from the solvent to the cavity forms in the $I_{0.36}$ state.

**Figure supplement 2.** Representative structures of the cavity region in the $I_{0.36}$ state.

**Figure supplement 3.** Tunnel clustering analysis on $I_{0.36}$ state.

**Figure supplement 4.** Ligand unbinding pathways revealed by ABMD simulations.

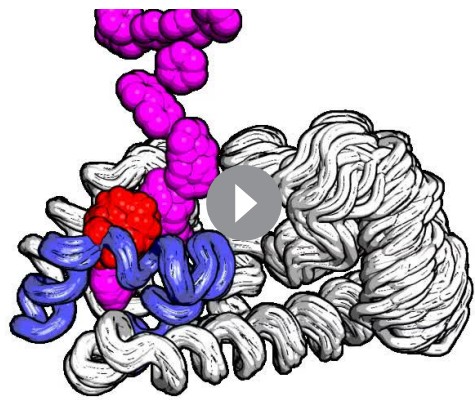

**Video 6.** A typical trajectory of the benzene escaping from the buried cavity of L99A via tunnel #1 revealed by ABMD simulations. The backbone of L99A is represented by white ribbons, Helices E, F and G are highlighted in blue, while F114 and benzene are represented by spheres in red and magenta, respectively.

different possible mechanism involves transient partial-loosening of the protein. In both cases, the mechanism differs from the reference path that we used as a guide to map the free energy landscape, demonstrating that high accuracy of the initial guess for a pathway is not absolutely required in the metadynamics simulations, suggesting also the more general applicability of the approach.

Finally, we observed a set of conformations with a transiently opened tunnel that leads from the exterior of the protein to the internal cavity, that is similar to a recently discovered path that is exposed when the cavity is filled by ligands of increasing size. The fact that such a tunnel can be explored even in the absence of ligands suggests that intrinsic protein motions may play an important role in ligand binding, and indeed we observed this path to be dominant in simulations of ligand unbinding.

In total, we present a global view of the many, sometimes coupled, dynamical processes present in a protein. Comparison with a range of experimental observations suggests that the simulations provide a relatively accurate description of the protein, demonstrating how NMR experiments can be used to benchmark quantitatively the ability of simulations to study conformational exchange. We envisage that future studies of this kind, also when less is known about the structure of the alternative states, will help pave the way for using simulations to study the structural dynamics of proteins and how this relates to function.

## Materials and methods

### System preparation

Our simulations were initiated in the experimentally determined structures of the ground state of L99A ($G_{Xray}$; PDB ID code 3DMV) or minor, E state ($E_{ROSETTA}$; 2LCB). The structure of the ground state of the L99A, G113A, R119P triple mutant, corresponding to the E state of L99A was taken from PDB entry 2LC9 ($G_{ROSETTA}^{Triple}$). Details can be found in the Appendix.

### Initial reaction path

Taking $G_{Xray}$ and $E_{ROSETTA}$ as the models of the initial and final structures, we calculated an initial reaction path between them with the MOIL software (*Elber et al., 1995*), which has been used to explore the mechanism of conformational change of proteins (*Wang et al., 2011*). Further details can be found in the Appendix and in refs. (*Majek et al., 2008*; *Wang et al., 2011*).

### Path CV driven metadynamics simulations with adaptive hills

The adaptive-hill version of metadynamics updates the Gaussian width on the fly according to the local properties of the underlying free-energy surface on the basis of local diffusivity of the CVs or the local geometrical properties. Here, we used the former strategy. Simulation were performed using Gromacs4.6 (*Pronk et al., 2013*) with the PLUMED2.1 plugin (*Tribello et al., 2014*). See parameter details in *Appendix 1—table 1*.

### Replica-averaged CS-restrained simulations

We performed replica-averaged CS restrained MD simulations by using GPU version of Gromacs5 with the PLUMED2.1 and ALMOST2.1 (*Fu et al., 2014*) plugins. Equilibrated structures of $E_{ROSETTA}$

and $G_{ROSETTA}^{Triple}$ were used as the starting conformations. CS data of $E_{ROSETTA}$ and $G_{ROSETTA}^{Triple}$ were obtained from the BMRB database (*Ulrich et al., 2008*) as entries 17604 and 17603, respectively.

## Reconnaissance metadynamics simulations

Reconnaissance metadynamics (*Tribello et al., 2010*) uses a combination of a machine learning technique to automatically identify the locations of free energy minima by periodically clustering the trajectory and a dimensional reduction technique that can reduce the landscape complexity. We performed several reconnaissance metadynamics simulations with different combinations of CVs starting from $G_{Xray}$ using Gromacs4.5.5 with PLUMED1.3 plugin. See parameter details in *Appendix 1—table 1*.

## Calculating kinetics using infrequent metadynamics

The key idea of infrequent metadynamics is to bias the system with a frequency slower than the barrier crossing time but faster than the slow intra-basin relaxation time, so that the transition state region has a low risk of being substantially biased. As the first transition times should obey Poisson statistics, the reliability of the kinetics estimated from InMetaD can be assessed by a statistical analysis based on the Kolmogorov-Smirnov (KS) test (*Salvalaglio et al., 2014*). See parameter details on Appendix and *Appendix 1—table 1*.

## Acknowledgements

We would like to thank Pratyush Tiwary and Matteo Salvalaglio for the help on the usage and analysis of InMetaD, Gareth Tribello for the help on the usage of reconnaissance metadynamics, Lewis E. Kay, Pengfei Tian, Shuguang Yuan, Micha Ben Achim Kunze for their helpful discussions and comments. Ludovico Sutto and Francesco Gervasio are thanked for input on metadynamics simulations.

## Additional information

### Funding

| Funder | Author |
| --- | --- |
| Novo Nordisk Foundation | Kresten Lindorff-Larsen |
| Carlsbergfondet | Kresten Lindorff-Larsen |
| Danish e-Infrastructure Cooperation | Kresten Lindorff-Larsen |

The funders had no role in study design, data collection and interpretation, or the decision to submit the work for publication.

### Author contributions

YW, Conception and design, Acquisition of data, Analysis and interpretation of data, Drafting or revising the article; EP, Acquisition of data, Analysis and interpretation of data; KL-L, Conception and design, Analysis and interpretation of data, Drafting or revising the article

### Author ORCIDs

Yong Wang, http://orcid.org/0000-0001-9156-0377
Elena Papaleo, http://orcid.org/0000-0002-7376-5894
Kresten Lindorff-Larsen, http://orcid.org/0000-0002-4750-6039

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

## Appendix

## Molecular modeling and system preparation

We used the crystal structure of T4L L99A (PDB ID code 3DMV) as starting point for simulations of the G state of L99A. For the E state of L99A and the G state of the L99A, G113A, R119P triple-mutant we used the CS-ROSETTA structures with PDB ID code 2LCB and 2LC9, respectively.

Each protein was solvated in a dodecahedral box of TIP3P water molecules with periodic boundary conditions. The protein-solvent box had a distance of 10 Å from the solute to the box boundary in each dimension, which results in approximately 10,000 water molecules and more than 32,000 atoms. Chloride counter-ions were included to neutralize the overall electric charge of the system. We used the CHARMM22* force field (*Piana et al., 2011*) for most of our simulations, but also used the Amber ff99SB*-ILDN (*Hornak et al., 2006*; *Best and Hummer, 2009*; *Lindorff-Larsen et al., 2010*) for some simulations to examine the dependency of the results on the choice of force field.

The van der Waals interactions were smoothly shifted to zero between 0.8 and 1.0 nm, and the long-range electrostatic interactions were calculated by means of the particle mesh Ewald (PME) algorithm with a 0.12 nm mesh spacing combined with a switch function for the direct space between 0.8 and 1.0 nm. The bonds involving hydrogen atoms were constrained using the LINCS algorithm. We employed the V-rescale thermostat (*Bussi et al., 2007*) to control the temperatureand simulated the system in the canonical ensemble.

## Reference transition path and path collective variables

We used path collective variables both to enhance sampling in path driven metadynamics (see below) as well as to represent the conformational landscape sampled by other means. Path collective variables have previously been shown to be very useful in finding free energy channels connecting two metastable states, and also able to construct the global free energy surfaces even far away from the initial path (*Branduardi et al., 2007*). A reference path is defined by a set of conformations along the path, and the progress along this path can be described mathematically as:

$$S_{path}(X) = \frac{\sum_{i=1}^{N} i e^{-\lambda M_i(X)}}{\sum_{i=1}^{N} e^{-\lambda M_i(X)}}$$

Here X are the coordinates of the instantaneous protein conformer sampled by MD simulations, N is the number of frames used to describe the reference path (often dependent on the length scale of the conformational transition process), $M_i(X)$ is the mean-square deviation (after optimal alignment) of a subset of the atoms from the reference structure of $i$ th frame, and $\lambda$ is a smoothing parameter whose value should roughly be proportional to the inverse of the average mean square displacement between two successive frames along the reference path. With this definition, $S_{path}$ quantifies how far the instantaneous conformer, $X$, is from the reactant state and the product state, thus monitoring the progress of the system along the conformational transition channel.

Using $S_{path}$ as the sole CV would assume that the initial reference path contains a sufficient description of the important degrees of freedom between the two states. It is, however, rarely possible to guess such a path because determining the actual pathway taken is a goal of the simulation. Thus, $S_{path}$ is supplemented by a second CV, $Z_{path}$, which measures the deviation away from the structures on the reference path. I.e. if $S_{path}$ quantifies the progress along the path, $Z_{path}$ measures the distance away from the reference path:

$$Z_{path}(X) = -\frac{1}{\lambda} ln \sum_{i=1}^{N} e^{-\lambda M_i(X)}$$

The combination of $S_{path}$ and $Z_{path}$ thus maps the entire conformational landscape to a two-dimensional projection, which can also be thought of as a tube connecting the two end states, and where $S$ measures the progress along the tube and $Z$ the width of the tube. The usefulness of the path CVs is, however, dependent on the quality of reference path, which is determined amongst other things by two factors: (1) the relative accuracy of the reference path and (2) how uniform reference structures are distributed along the path. Because of the explosion in number of possible conformations as one progresses along $Z_{path}$, simulations are mostly enhanced when $S_{path}$ provides a relatively good description of the pathway taken. Further, if reference structures are only placed sparsely along the path, one looses resolution of the free energy surface and also decreases the ability to enhance sampling.

To obtain a good reference path, without knowing beforehand the mechanism of conversion, we here used a method to construct snapshots along a possible initial path. Taking $G_{Xray}$ and $E_{ROSETTA}$ as initial and final structures, we calculated the optimal reaction paths between them with the MOIL software (**Elber et al., 1995**) which has previously been used to explore the mechanism of conformational change of proteins (**Wang et al., 2011**) . After minimizing endpoint structures, we employed the minimum-energy-path self-penalty walk (SPW) (**Czerminski and Elber, 1990**) functional embedded in the CHMIN module to obtain an initial guess for the conformational transition path. This path was subsequently optimized in the SDP (steepest descent path) module by minimizing the target function $T$ consisting of two terms $S$ and $C$. $S$ is an action function that provides approximate most probable Brownian trajectories connecting the reactant and product states, while $C$ is a restraint function aimed to distribute frames approximately uniformly along the path. They can be expressed by:

$$S = \sum_{i=1}^{N-1} \sqrt{H_s + (\frac{\partial U}{\partial x_i})^2} |x_{i+1} - x_i|$$
$$C = \lambda \sum_{i} (\Delta l_{i,i+1} - <\Delta l>)^2$$

where $N$ is the number of frames along the reference path, $x_i$ is the entire vector of conformational coordinates of frame $i$, $U$ is a potential energy as a function of the mass-weighted coordinate vector, $H_s$ is a constant with an arbitrary positive value, which can be tuned to generate the optimal paths with different thermal energies. $\Delta l_{i,i+1} = M^{1/2} |x_i - x_{i+1}|$ is the arc-length between consecutive frames. $\lambda$ controls the strength of the restraint function $C$.

The SDP or minimum energy path is the limiting path in which the action $S$ is optimized with $H_s \rightarrow 0$. An important advantage of an SDP is that it is capable of giving a good guess of the minimal energy path which can reflect the major mechanism, only with inexpensive computation. Further details can be found in Refs. (**Wang et al., 2011**; **Majek et al., 2008**).

The SDP was approximated as 31 discrete conformations. Most regions of the $G_{Xray}$ and $E_{ROSETTA}$ structures are very similar, with the exception of the cavity-related atoms whose movement determines the conformational transitions between the G and E states. To minimize computations, we used only a subset of heavy atoms around the cavity from amino acid residues 99 to 126 to define $S_{path}$ and $Z_{path}$, resulting a 'light version' of the reference path which included only 212 atoms. We used the $C_\alpha$ atoms of the C-terminal domain to align the molecule. It is important to note that by focusing only on atoms surrounding the cavity in the calculation of the path-variables we only enhance sampling of the conformational changes relating to the cavity. We used $\lambda = 56.0$ based on the consideration that smooth change of the function of $S_{path}$ can be achieved when $e^{-\lambda <M_i(X)>} \geq 0.1$ (**Tribello et al., 2014**).

We defined $S_{path}$ and $Z_{path}$ using the SDP as described above. The equidistant requirement of the path is satisfied by the penalty function as is evident from the RMSD matrix for path frames which has a gullwing shape (*Figure 2—figure supplement 1*, indicating that each frame is closest to its neighbor and more different from all other reference frames.

## Metadynamics simulations

Metadynamics discourages the system from sampling already visited conformational regions by continuously adding an external history-dependent repulsive potential at the present value of the reaction coordinates or CVs, which are assumed to include the slowly varying degrees of freedom and thus describe the main features of the dynamics (*Laio and Parrinello, 2002*). The biasing potential in metadynamics results in an artificial (enhanced) dynamics but makes it possible to reconstruct the free energy surfaces by removing the bias introduced. The bias is typically added as Gaussians at regular time intervals, $\tau_G$, and is given by:

$$V_G(S,t) = \sum_{k=1}^{|t/\tau_G|} \omega_k \delta(S(t), S(k \cdot \tau_G))$$

Here $S$ denotes the CVs, $k$ is the index of the individual Gaussians, $\omega_k$ is the height of the $k$'th Gaussian, and $\delta(S(t), S(k \cdot \tau_G))$ is a short-ranged kernel function of the CVs:

$$\delta(S(t), S(k \cdot \tau_G)) = e^{\sum_{j=1}^{n} -\frac{|s_j(t) - s_j(k \cdot \tau_G)|^2}{2\sigma_{kj}^2}}$$

where $n$ is the number of CV, $j$ is the index of a CV, and $\sigma_{kj}$ and $s_j(k \cdot \tau_G)$ are the width and the position of the Gaussian hills, respectively.

We here used a range of different metadynamics approaches to determine the free energy landscape, and the mechanism and kinetics of conformational exchange (see below).

## Well-tempered metadynamics

In the well-tempered version of metadynamics, the height of the individual Gaussians, $\omega_k$, is decreased as the total bias accumulates over time, in order to improve the convergence of the free energy:

$$\omega_k = \omega_0 e^{-\frac{1}{\gamma-1} \frac{V_G(S,k \cdot \tau_G)}{k_B T}}$$

Here $\omega_0$ is the initial height, $\gamma = (T + \Delta T)/T$ is referred as the bias factor, which can be tuned to control the speed of convergence and diminish the time spent in lesser-relevant, high-energy states. Thus, the quantity $T + \Delta T$ is often referred as the fictitious CV temperature.

## Adaptive-width metadynamics

In contrast to standard metadynamics in which the width of the Gaussians is constant, the adaptive-width version of metadynamics updates the Gaussian width $\sigma_{kj}$ on the fly according to local properties of the underlying free-energy surface on the basis of local diffusivity of the CVs or the local geometrical properties (*Branduardi et al., 2012*).

In the region of conformational space near the endpoints of the path CVs many conformations are compressed on similar CV values, leading to high-density but low-fluctuation boundaries. It is apparent that the use of a fixed width might give an inaccurate estimation of the free energy profile in the boundaries where the free energy basins of reactant and product states

are located, also makes it more difficult for the simulations to converge. Therefore, the feature of shape adaptive of Gaussian potential is particularly helpful for the case of using path CVs that have significant boundary effects.

## Metadynamics with path variables (PathMetaD)

We sampled the free energy landscape along $S_{path}$ and $Z_{path}$, as defined above, using adaptive-width metadynamics, which resulted in a finer resolution and faster convergence of the free energy landscape, in particular near the path boundaries, than standard metadynamics. The production simulations were performed at 298K in well-tempered ensemble (*Appendix 1— table 1*: RUN1 and RUN2).

## Reconnaissance metadynamics

To explore the mechanism of conformational exchange we used reconnaissance metadynamics (*Tribello et al., 2010*). This is a 'self-learning' approach which combines of a machine learning technique to automatically identify the locations of free energy minima by periodically clustering the trajectory and a dimensional reduction technique that can reduce the complex locations to a locally-defined one-dimensional CV by using information collected during the clustering. It has previously been shown that reconnaissance metadynamics makes it possible to determine a path from a large set of input collective variables (*Söderhjelm et al., 2012*; *Tribello et al., 2011*).

## Infrequent Metadynamics (InMetaD)

We used the recently described 'infrequent metadynamics' (InMetaD) to obtain the rates of the conformational exchange process (*Tiwary and Parrinello, 2013*). In standard metadynamics simulations it is very difficult to obtain kinetic properties because the biasing potential is added both to the free energy basins as well as the barriers that separate them. While it is potentially possible to determine the rates from the height of the free energy barriers, this requires both that the CVs used represent the entire set of slowly varying degrees of freedom, and also a good estimate of the pre-exponential factor to convert barrier height to a rate.

The key idea in InMetaD to circumvent these problems is to attempt to add the bias to the system more slowly than the barrier crossing time but faster than the slow inter-basin relaxation time, so that the transition state region has a lower risk of being biased, and therefore the transitions are less affected. By filling up a free energy basin by a known amount it is possible to determine how much the barrier has been decreased, and hence remove this bias from the rates determined. Thus, as described in more detail below, the approach works by performing a large number of individual simulations to obtain first passage times between the individual basins, which are then corrected by the known enhancement factors to obtain estimates of the unbiased rates. This method has been successfully used to reproduce the kinetics of conformational change of alanine dipeptide (*Tiwary and Parrinello, 2013*) unbinding of the inhibitor benzamidine from trypsin (*Tiwary et al., 2015a*) and slow unbinding of a simple hydrophobic cavity-ligand model system (*Tiwary et al., 2015b*).

In these simulations we used a deposition frequency of 80 or 100 ps (see parameters in *Appendix 1—table 1*), a value much lower than the deposition frequency of 1 ps used in the PathMetaD simulations described above. In this way we lower the risk of substantially corrupting the transition state region. In addition, a tight upper wall potential on $Z_{path}$=0.10 nm$^2$ is used to confine the sampling based on our converged free energy surface which shows the conformational change mostly occurs within this region.

With these parameters we collected dozens of trajectories that have a state-to-state transition in the G-to-E and E-to-G directions. The passage times observed in each of these were then corrected for the metadynamics bias as follows.

First, we calculate the acceleration factor $\alpha$ from:

$$\alpha = \tau/\tau_M = <e^{V(s,t)/kT}>_M$$

where the anguluar brackets denote an average over a metadynamics run before the first transition, and $V(s,t)$ is the metadynamics time-depedent bias. The evolution of the acceleration factor $\alpha(t)$ can be expressed by:

$$\alpha(t) = (1/t) \int_0^t dt' e^{V(s,t')/kT}$$

Then the observed passage time, $t$, is reweighted by:

$$\tau_{true} = \alpha(t) * t = \int_0^t dt' e^{V(s,t')/kT}$$

In principle, the transition time should be a Poisson-distributed random variable, and its mean, $\mu$, standard deviation $\sigma$ and median $t_m/ln2$ all should be equal to each other. In practice, however, they are somewhat sensitive to insufficient sampling (*Salvalaglio et al., 2014*). So rather than simply calculating averages of the individual times, we estimated the average rate and transition time $\tau$ from a fit of the empirical cumulative distribution function (ECDF) with the theoretical cumulative distribution function (TCDF):

$$TCDF = 1 - e^{-\frac{t}{\tau}}$$

It has previously been shown that $\tau$ estimated in this way converges more quickly than the simple average, $\mu$. This is also consistent with our observation, and we find that 10–15 samples appear sufficient to get a reasonably accurate estimation of the transition time. We used a bootstrap approach to estimate the errors.

To examine whether the observed times indeed follow the expected Poisson distribution we used a Kolmogorov-Smirnov (KS) test to obtain a p-value that quantifies the similarity between the empirical and theoretical distributions. Traditionally, a threshold value typically of 0.05 or 0.01 (the significance level of the test) is used to judge if the theoretical (TCDF) and empirical (ECDF) distributions are in agreement. If the p-value is equal to or larger than the threshold value, it suggests that the estimated transition time is quite reliable. If (a) the transition regions were perturbed significantly with infrequent biasing or (b) there are hidden unidentified timescales at play (e.g. the CVs do not capture the slow degrees of freedom) the KS test for time-homogeneous Poisson statistics would fail.

## CS-restrained replica-averaged simulation

The simulation methods described above constitute different ways of exploring the thermodynamics (PathMetaD), kinetics (InMetaD) and mechanism (Reconnaissance metadynamics) of conformational exchange. In all of these simulations, sampling is determined by the molecular energy function (force field), and the experimental information on T4L is used only in the construction of the path variables. When additional experimental information is available, one may introduce an additional energy term so as to bias the simulations to be in agreement with this information (*Robustelli et al., 2010*). As the experimental values are ensemble averages we apply these restraints only to averages calculated over a number of 'replicas' that are simulated in parallel. In this way, the information from the experimental data is incorporated into the simulation as a perturbation following the maximum entropy principle (*Pitera and Chodera, 2012*; *Roux and Weare, 2013*; *Cavalli et al., 2013*; *Boomsma et al., 2014*).

We used this approach to obtain conformational ensembles that include not only information from the molecular force field, but also experimental NMR chemical shifts (CS). In particular, we used either the chemical shifts of the E state of L99A obtained from the analysis of the CPMG experiments, or the native state chemical shifts of a triple mutant that populates the same state as its ground state (**Bouvignies et al., 2011**). The CS restraints were imposed by adding a pseudo-energy term ($E_{CS}$) to a standard molecular-mechanics force field ($E_{FF}$).

$$E_{CS} = \epsilon_{CS} \sum_{i=1}^{N} \sum_{j=1}^{6} (\delta_{ij}^{Exp} - \frac{1}{M} \sum_{k=1}^{M} \delta_{ijk}^{Sim})^2$$

Here $\epsilon_{CS}$ is strength of the CS restraints, $i$ indicates the residue number (total of $N$), $j$ indicates each of the six backbone atoms whose chemical shifts were used ($C_\alpha$, $C_\beta$, $C'$, $H_\alpha$, HN and N), $k$ is an index for the total of $M$ replicas, and $\delta^{Exp}$ and $\delta^{Sim}$ are the experimental and simulated CSs, respectively. The latter quantity, $\delta^{Sim}$, was calculated by CamShift (**Fu et al., 2014**) as a plugin of PLUMED. The CS values of Pro, Gly, Asp, Glu, His and terminal residues were not included because the accuracy in their predictions are too low to contain sufficient information in this approach (**Fu et al., 2014**). We set $\epsilon_{CS}$=24 kJ mol$^{-1}$ ppm$^{-2}$ and used M=4 replicas. In principle, the number of replicas is a free parameter that should be set as large as possible when the experimental data and the method for calculating it is noise free (**Boomsma et al., 2014**). In practice, one uses a finite set of replicas and it has been shown that M=4 replicas is sufficient to capture the dynamics accurately (**Camilloni et al., 2013**).

We performed replica-averaged CS restrained MD simulations using GROMACS4.6 and the PLUMED2.1 plugin at 298 K. The equilibrated structures of $E_{ROSETTA}$ and $G_{ROSETTA}^{Triple}$ were used as the starting configuration (of each of the four replicas) in the CS-restrained simulations of L99A and the L99A,G113A,R119P triple mutant, respectively. The CS data for $E_{ROSETTA}$ and $G_{ROSETTA}^{Triple}$ were obtained from the Biological Magnetic Resonance Bank (BMRB) database (**Ulrich et al., 2008**) with entries 17604 and 17603, respectively.

## PT-WT-MetaD failed to get the converged free energy landscape

In practice, the choice of CVs plays a fundamental role in determining the accuracy, convergence and efficiency of metadynamics simulations. If an important CV is missing, the exploration of the free energy landscape will be difficult due to hysteresis. Finding a minimal set of CVs that include all important degrees of freedom is a highly nontrivial task and one often has to proceed by several rounds of trial simulations.

At the beginning, we followed a strategy which had previously been successfully used in the exploration of protein conformational transitions (**Sutto and Gervasio, 2013**; **Papaleo et al., 2014**) to design a set of CVs on the basis of static structural comparison between $G_{Xray}$ and $E_{ROSETTA}$. In particular, by comparing these two structures we defined several CVs that described structural differences by individual dihedral angles and hydrogen bonds, as well as dihedral correlation and coordination number (state-specific contact map) (**Bonomi et al., 2009**) (summarized in **Appendix 1—table 1**). We combined these in a multiple-replica, parallel tempering approach in the well-tempered ensemble (PT-WT-metaD) (**Bonomi and Parrinello, 2010**) to further enhance the sampling. In PT-WT-metaD, the energy fluctuations are enlarged by using energy as a biased CV but the average energy is the same as the canonical ensemble, allowing the use of a larger spacing between temperatures and a much fewer number of replicas than normal PT simulations (**Barducci et al., 2015**). Coordinate exchange with high temperature replicas can enhance the sampling of all the degrees of freedom, even those not included in the biased CVs, and one may include a 'neutral' replica (without energy bias, at 298 K). We performed a series of simulations with different combinations of CVs starting from the G state of L99A (**Appendix 1—table 2**). However, unfortunately, we only observed partial G-to-E transitions, even in a relatively long trajectory

of about 1 $\mu$s for each replica. This negative result suggested that these manually chosen CVs did not contain all the necessary slow degrees of freedom.

## Tunnel analysis

We used CAVER3 (*Chovancova et al., 2012*) to analyse the structures and CAVER Analyst1.0 (*Kozlikova et al., 2014*)(http://www.caver.cz/, see also parameters in *Appendix 1—table 4*) to separate the tunnels into different clusters. CAVER Analyst is a standalone program based on CAVER3.0 algorithm (*Chovancova et al., 2012*). The settings for the tunnel calculations can be set through the Tunnel Computation window, while the advanced parameters can be set in the Tunnel Advanced Settings window. We used the center of mass of the cavity-related region (residue 93-124) as the position of the starting point. Average-link hierarchical clustering algorithm is performed to build a tree hierarchy of tunnel axes based on their pairwise distances. The size of the resulting clusters is dependent on the Clustering threshold parameter which specifies the level of detail at which the tree hierarchy of tunnel clusters will be cut. We used the default value so that the tree hierarchy of tunnel clusters is cut at the value of 3.5.

## Adiabatic bias molecular dynamics

Adiabatic biased molecular dynamics (ABMD) (*Marchi and Ballone, 1999*; *Paci and Karplus, 1999*) is an algorithm developed to accelerate the transition from the reactant state to the productive state, here corresponding to the ligand bound state and ligand-free state, respectively. In ABMD the system is perturbed by a 'ratcheting potential', which acts to 'select' spontaneous fluctuations towards the ligand-free state. The ratcheting potential is implemented in PLUMED2.2 as

$$V(\rho(t)) = \begin{cases} 0.5K(\rho(t) - \rho_m(t))^2, & \rho(t) > \rho_m(t) \\ 0, & \rho(t) \leq \rho_m(t) \end{cases}$$

where

$$\rho(t) = (S(t) - S_{target})^2$$

and

$$\rho_m(t) = min_{0 \leq \tau \leq t} \rho(\tau) + \eta(t)$$

$K$ is the force constant, $S(t)$ is the instantaneous CV value, $S_{target}$ is the target value of the CV and $\eta(t)$ is an additional white noise acting on the minimum position of $\rho(t)$. Here, we used the RMSD of the ligand to the cavity-bound state as the CV, and set $S_{target} = 4.0$ nm and K=20 kJ $\cdot$ mol$^{-1}$ $\cdot$ nm$^{-2}$ or 50 kJ $\cdot$ mol$^{-1}$ $\cdot$ nm$^{-2}$ to check the dependency of the force constant chosen. The biasing potential is zero when the CV increases but provides a penalty when the CV decreases. In this way, we were able to observe multiple unbinding events in simulations despite the long lifetime (1.2 ms) of the ligand in the cavity.

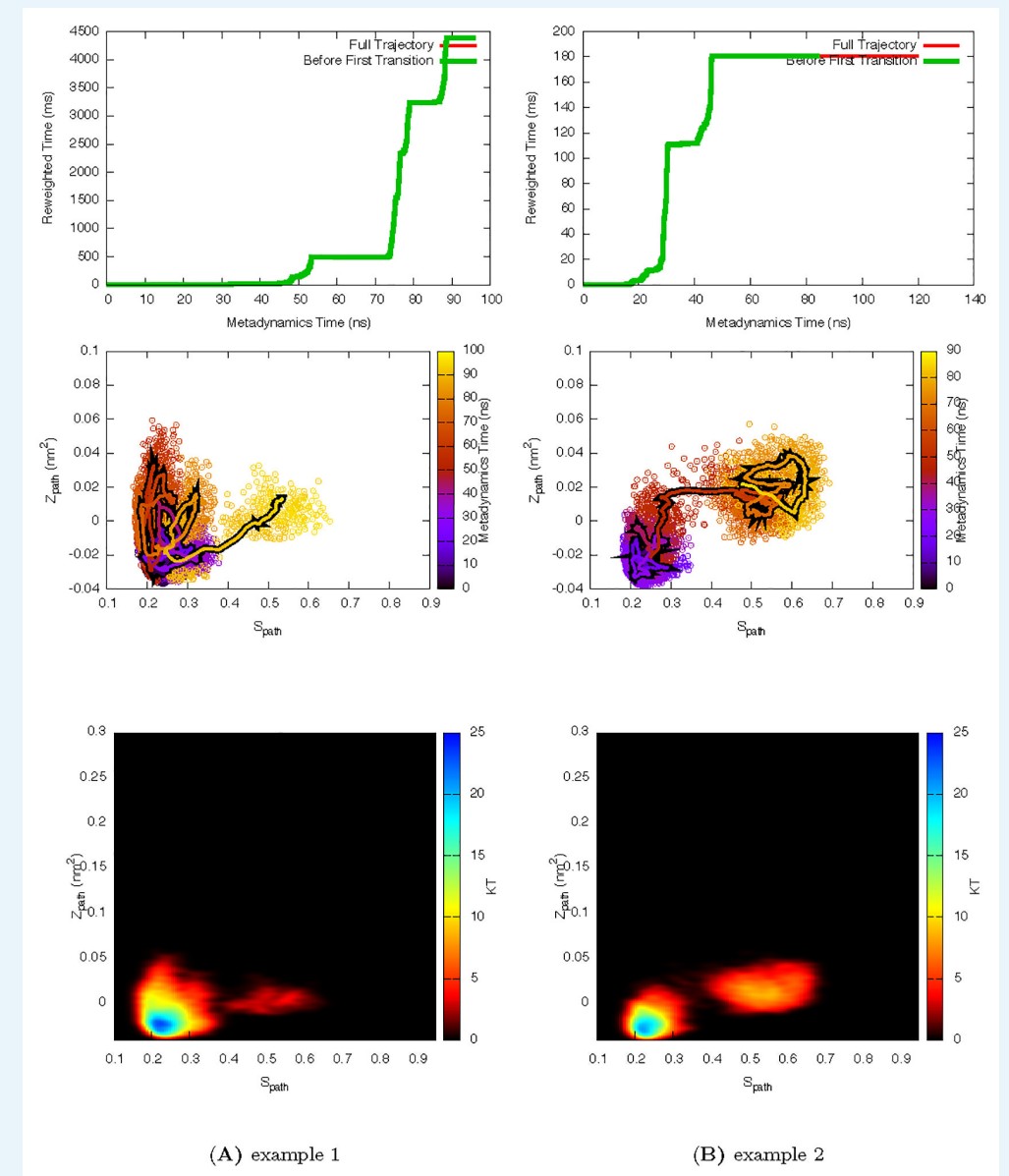

**Appendix 1—figure 1.** Two representative InMetaD trajectories of L99A with G to E transitions. The time point, t′, for the first transition from G to E is identified when the system evolves into conformational region of $S_{path}$ > 0.55 and $Z_{path}$ < 0.01. We then calculate the unbiased passage time by multiplying t′ by the corresponding accelerate factor α(t′). Upper panels show the evolution of the reweighted time as a function of metadynamics time. The kinks usually indicate a possible barrier crossing event. Middle panels show the trajectories starting from the G state and crossing the barrier towards the E state. Lower panels show the biasing landscape reconstructed from the deposited Gaussian potential, which can be used to check the extent to which the transition state regions are affected by deposited bias potential.

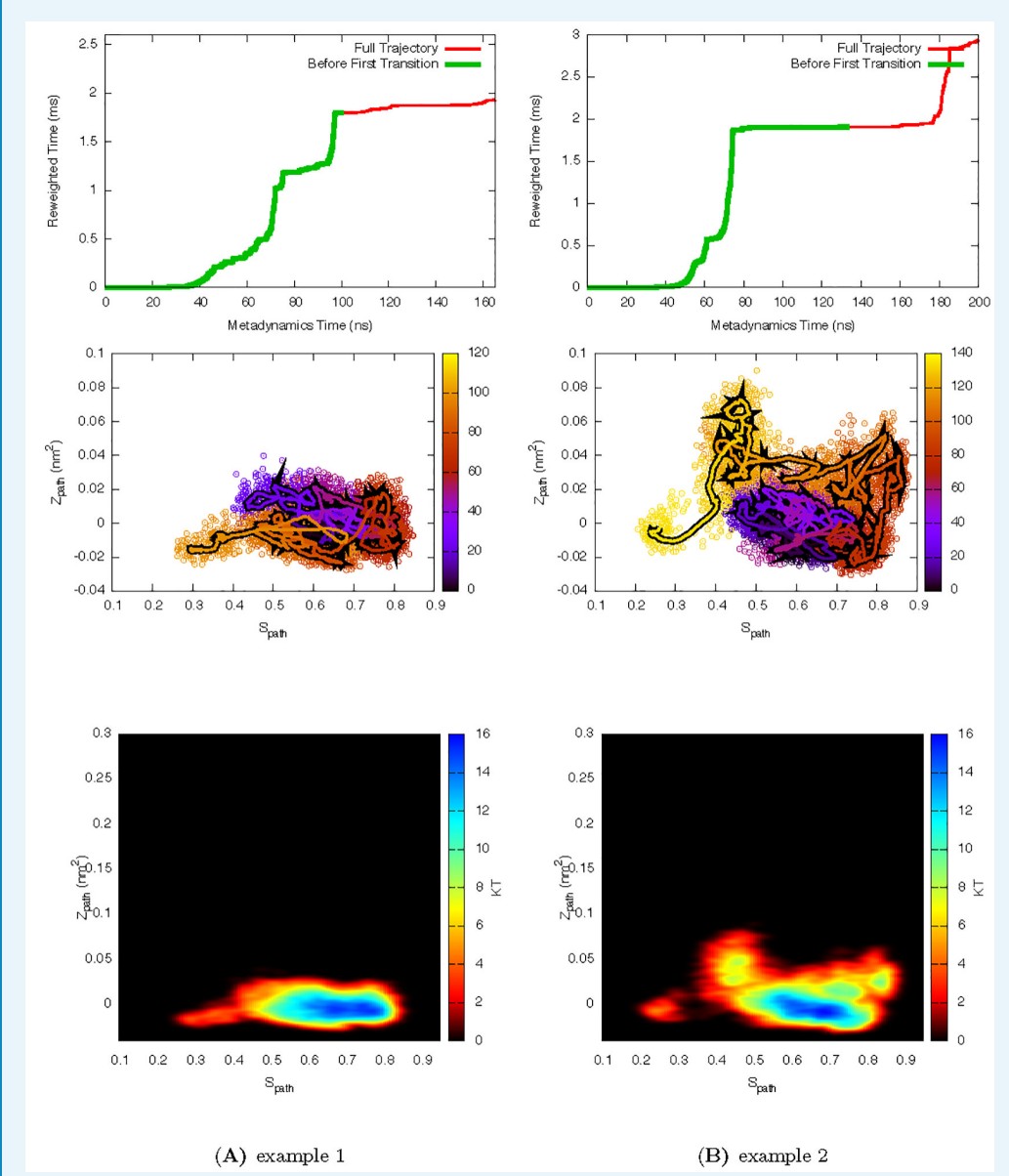

**Appendix 1—figure 2.** Two representive InMetaD trajectories of L99A with E to G transitions of L99A. First transition time for G to E transition is identified when the system evolves into conformational region of $S_{path} < 0.28$ and $Z_{path} < -0.01$.

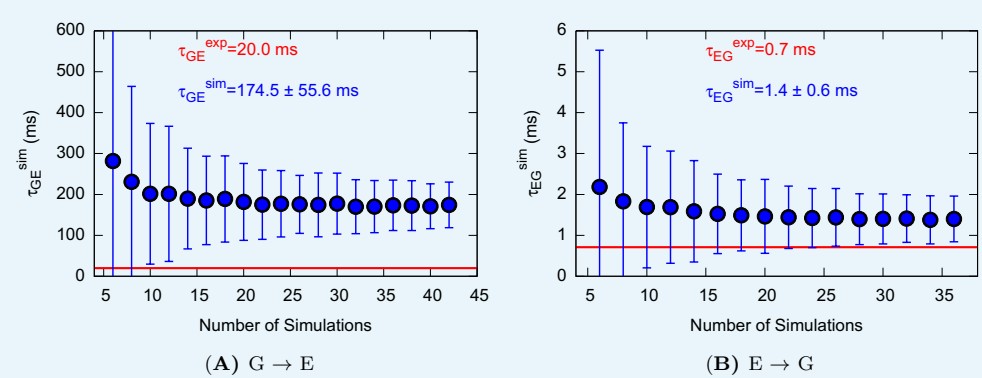

**(A)** G → E     **(B)** E → G

**Appendix 1—figure 3.** Characteristic transition times between G and E states of L99A. The error bars represent the standard deviation of τ obtained from a bootstrap analysis.

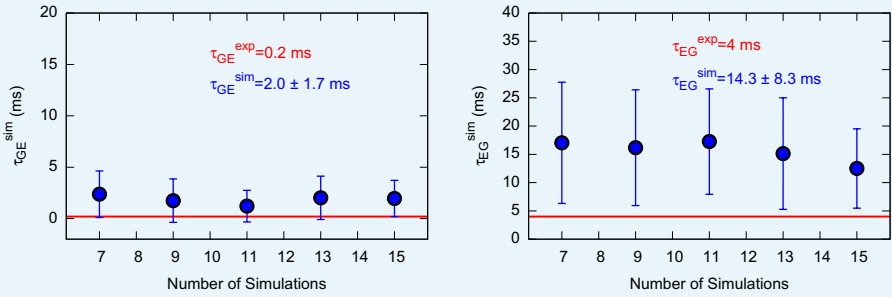

**Appendix 1—figure 4.** Characteristic transition times between G and E states of the triple mutant. The figure shows the characteristic transition time $\tau_{G \to E}$ (left panel) and $\tau_{E \to G}$ (right panel) of the triple mutant as a function of the size of a subsample of transition times randomly extracted from the main complete sample. The error bars represent the standard deviation of characteristic transition times obtained by a bootstrap analysis. The calculated and experimental values of the transition times are shown in blue and red font, respectively.

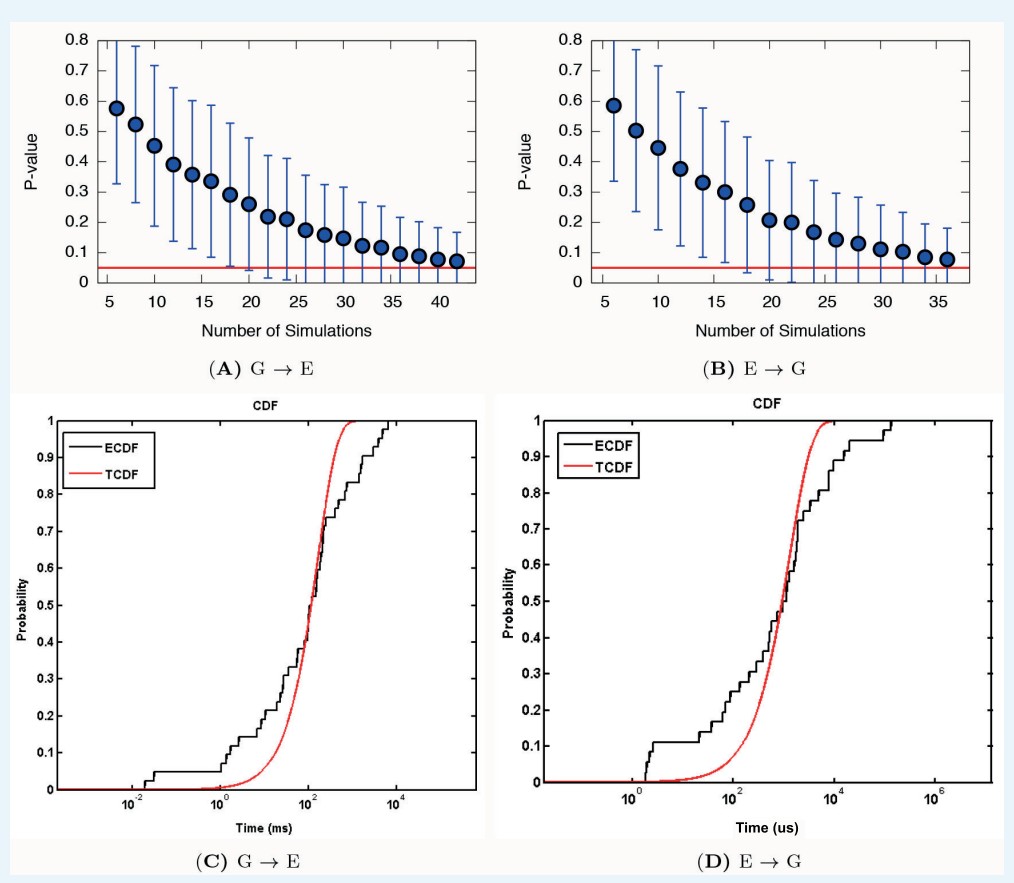

**Appendix 1—figure 5.** Poisson fit analysis for G to E transitions and E to G transitions of L99A. We show the ECDF (the empirical cumulative distribution function) and TCDF (the theoretical cumulative distribution function) in black and blue lines, respectively. The respective p-values are reasonably, albeit not perfectly, well above the statistical threshold of 0.05 or 0.01, indicating the kinetics is not substantially modified by the deposited bias potential in InMetaD. Error bars are the standard deviation obtained by a bootstrap analysis.

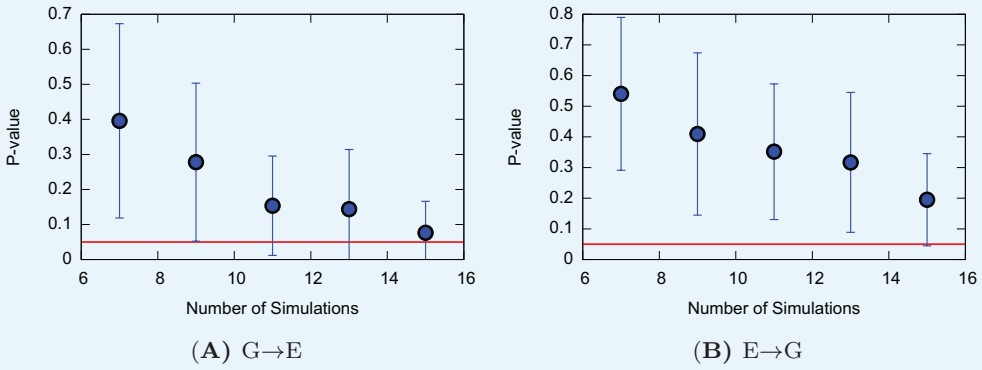

**Appendix 1—figure 6.** Poisson fit analysis for G to E transitions and E to G transitions of the triple mutant. The figure shows the p-values of the Poisson fit analysis of $G \rightarrow E$ (**A**) and $E \rightarrow G$ (**B**) transition times as a function of the size of a subsample of transition times randomly extracted from the main complete sample.

**Appendix 1—table 1.** Simulation details.

| Method | label | system | init | length | force field | CVs | parameters | T (K) | software version |
|---|---|---|---|---|---|---|---|---|---|
| PathMetaD | RUN1 | L99A | G | 667 ns | CHARMM22* | $S_{path}, Z_{path}$ | $\gamma=20^\dagger, \tau_D=1\ ps^\ddagger, \tau_G=1\ ps^\S, \omega_0=1.5^\P$ | 298 | PLUMED2.1, GMX4.6 |
| PathMetaD | RUN2 | L99A, G113A, R119P | G | 961 ns | CHARMM22* | $S_{path}, Z_{path}$ | $\gamma=20, \tau_D=1\ ps, \tau_G=1ps, \omega_0=1.5$ | 298 | PLUMED2.1, GMX5.0 |
| CS-restrained MD | RUN3 | L99A | E | 252 ns | CHARMM22* | | $N^\#=4, \epsilon_{CS}=24^{**}$ | 298 | PLUMED2.1, ALMOST2.1, GMX5 |
| CS-restrained MD | RUN4 | L99A | E | 233 ns | CHARMM22* | | $N=2, \epsilon_{CS}=24$ | 298 | PLUMED2.1, ALMOST2.1, GMX5 |
| CS-restrained MD | RUN5 | L99A | E | 221 ns | CHARMM22* | | $N=2, \epsilon_{CS}=12$ | 298 | PLUMED2.1, ALMOST2.1, GMX5 |
| CS-restrained MD | RUN6 | L99A | E | 125 ns | AMBER99sb*ILDN | | $N=4, \epsilon_{CS}=24$ | 298 | PLUMED2.1, ALMOST2.1, GMX5 |
| CS-restrained MD | RUN7 | L99A | E | 190 ns | AMBER99sb*ILDN | | $N=2, \epsilon_{CS}=12$ | 298 | PLUMED2.1, ALMOST2.1, GMX5 |
| CS-restrained MD | RUN8 | L99A, G113A, R119P | G | 204 ns | CHARMM22* | | $N=4, \epsilon_{CS}=24$ | 298 | PLUMED2.1, ALMOST2.1, GMX5 |
| CS-restrained MD | RUN9 | L99A, G113A, R119P | G | 200 ns | CHARMM22* | | $N=2, \epsilon_{CS}=24$ | 298 | PLUMED2.1, ALMOST2.1, GMX5 |
| Reconnaissance MetaD | RUN10 | L99A | G | 120 ns | CHARMM22* | 2,3,4,5 | $\Delta T=10000^{\dagger\dagger}, \Delta r=250$ | 298 | PLUMED1.3, GMX4.5 |
| Reconnaissance MetaD | RUN11 | L99A | G | 85 ns | CHARMM22* | 2,3,4,5 | 10000, 250 | 298 | PLUMED1.3, GMX4.5 |
| Reconnaissance MetaD | RUN12 | L99A | G | 41 ns | CHARMM22* | 2,3,4,5,7,8 | 100000, 500 | 298 | PLUMED1.3, GMX4.5 |
| PT-WT-MetaD | RUN13 | L99A | G | 961 ns | CHARMM22* | 1,2,3,7,8 | $\gamma=20, \omega_0=0.5$ | 297 298‡‡ 303 308 316 325 341 350 362 | PLUMED 1.3, GMX4.5 |
| PT-WT-MetaD | RUN14 | L99A | G | 404 ns | CHARMM22* | 1,2,3,6,7 | $\gamma=20, \omega_0=0.5$ | 297 298 303 308 316 325 333 341 350 | PLUMED 1.3, GMX4.5 |

*Appendix 1—table 1 continued on next page*

*Appendix 1—table 1 continued*

| Method | label | system | init | length | force field | CVs | parameters | T (K) | software version |
|---|---|---|---|---|---|---|---|---|---|
| PT-WT-MetaD | RUN15 | L99A | G | 201 ns | CHARMM22* | 1,2,3,6 | $\gamma=20,\ \omega_0=0.5$ | 297 298 303 308 316 325 333 341 350 | PLUMED 1.3, GMX4.5 |
| PT-WT-MetaD | RUN16 | L99A | G | 511 ns | CHARMM22* | 1,2,3,4,5 | $\gamma=5,\ \omega_0=0.5$ | 295 298 310 325 341 358 376 | PLUMED 1.3, GMX4.5 |
| PT-WT-MetaD | RUN17 | L99A | G | 383 ns | CHARMM22* | 1,2,3,4,5 | $\gamma=20,\ \omega_0=0.5$ | 298 305 313 322 332 337 343 | PLUMED 1.3, GMX4.5 |
| PT-WT-MetaD | RUN18 | L99A | E | 365 ns | CHARMM22* | 1 | $\gamma=20,\ \omega_0=0.5$ | 298 302 306 311 317 323 330 338 | PLUMED 1.3, GMX4.5 |
| plain MD | RUN19 | L99A | G | 400 ns | CHARMM22* | | | 298 | GMX5.0 |
| plain MD | RUN20 | L99A | E | 400 ns | CHARMM22* | | | 298 | GMX5.0 |
| plain MD | RUN21 | L99A | G | 400 ns | AMBER99sb*ILDN | | | 298 | GMX5.0 |
| plain MD | RUN22 | L99A | E | 400 ns | AMBER99sb*ILDN | | | 298 | GMX5.0 |
| plain MD | RUN23 | L99A,G113A,R119P | G | 400 ns | CHARMM22* | | | 298 | GMX5.0 |
| plain MD | RUN24 | L99A,G113A,R119P | E | 400 ns | CHARMM22* | | | 298 | GMX5.0 |
| plain MD | RUN25 | L99A,G113A,R119P | G | 400 ns | AMBER99sb*ILDN | | | 298 | GMX5.0 |
| plain MD | RUN26 | L99A,G113A,R119P | E | 400 ns | AMBER99sb*ILDN | | | 298 | GMX5.0 |
| InMetaD | RUN27-68[§§] | L99A | G | | CHARMM22* | $S_{path}, Z_{path}$ | $\gamma=20,\ \tau_G=80$ ps, $\omega_0=1.0,\ \sigma_S=0.016$[¶¶], $\sigma_Z=0.002$[##] | 298 | PLUMED2.1, GMX5 |
| InMetaD | RUN69-104[***] | L99A | E | | CHARMM22* | $S_{path}, Z_{path}$ | $\gamma=20,\ \tau_G=80$ ps, $\omega_0=1.0,\ \sigma_S=0.016,\ \sigma_Z=0.002$ | 298 | PLUMED2.1, GMX5 |
| InMetaD | RUN105-119 | L99A,G113A,R119P | G | | CHARMM22* | $S_{path}, Z_{path}$ | $\gamma=15,\ \tau_G=100$ps, $\omega_0=0.8,\ \sigma_S=0.016$[¶¶], $\sigma_Z=0.002$[##] | 298 | PLUMED2.2, GMX5.1.2 |
| InMetaD | RUN120-134 | L99A,G113A,R119P | E | | CHARMM22* | $S_{path}, Z_{path}$ | $\gamma=15,\ \tau_G=100$ ps, $\omega_0=0.8,\ \sigma_S=0.016$[¶¶], $\sigma_Z=0.002$[##] | 298 | PLUMED2.2, GMX5.1.2 |

Appendix 1—table 1 continued

| Method | label | system | init | length | force field | CVs | parameters | T (K) | software version |
|---|---|---|---|---|---|---|---|---|---|
| ABMD | RUN135-154 | L99A-Benzene | bound | | CHARMM22* | $RMSD_{benzene}$ | $K=50$ KJ $\cdot$ mol$^{-1}$ $\cdot$ nm$^{-2}$, $S_{target} = 4.0$ nm | 298 | PLUMED2.2, GMX5.1.2 |
| ABMD | RUN155-174 | L99A-Benzene | bound | | CHARMM22* | $RMSD_{benzene}$ | $K=20$ KJ $\cdot$ mol$^{-1}$ $\cdot$ nm$^{-2}$, $S_{target} = 4.0$ nm | 298 | PLUMED2.2, GMX5.1.2 |

† $\gamma$ is the bias factor.

‡ $\tau_D$ is the characteristic decay time used for the dynamically-adapted Gaussian potential.

§ $\tau_G$ is the deposition frequency of Gaussian potential.

¶ $\omega_0$ is the height of the deposited Gaussian potential (in KJ $\cdot$ mol$^{-1}$).

# N is the number of replicas.

** $\epsilon_{CS}$ is the strength of chemical shift restraints (in KJ $\cdot$ mol$^{-1}$ $\cdot$ ppm$^{-2}$).

†† $\Delta T$ means RUN_FREQ and $\Delta t$ means STORE_FREQ. Other parameters: BASIN_TOL=0.2, EXPAND_PARAM=0.3, INITIAL_SIZE=3.0.

‡‡ Replica at 298 K is the neutral replica without energy bias.

§§ 42 trajectories collected for G-to-E transitions.

¶¶ $\sigma_S$ is the Gaussian width of $S_{path}$.

## $\sigma_Z$ is the Gaussian width of $Z_{path}$ (in nm$^2$).

*** 36 trajectories collected for E-to-G transitions.

**Appendix 1—table 2.** Definition of collective variables.

| CV | definitions | parameters | purpose |
|---|---|---|---|
| 1 | total energy | bin=500 | enhance energy fluctuations |
| 2 | dihedral angle of $C_\alpha$ atoms of consecutive residues F104-Q105-M106-G107 | $\sigma$=0.1 | |
| 3 | dihedral angle of $C_\alpha$ atoms of consecutive residues G113-F114-T115-N116 | $\sigma$=0.1 | |
| 4 | $Q_G$, distance in contact map space to the $G_{Xray}$ structure | $\sigma$=0.5 | |
| 5 | $Q_E$, distance in contact map space to the $E_{ROSETTA}$ structure | $\sigma$=0.5 | |
| 6 | distance between $Q_G$ and $Q_E$ | $\sigma$=0.5 | |
| 7 | number of backbone hydrogen bonds formed between M102 and G107 | $\sigma$=0.1 | |
| 8 | dihedral correlation between the $C_\alpha$ dihedral angles of consecutive residues in segment N101-G107 | $\sigma$=0.1 | |
| 9 | global RMSD to the whole protein | wall potential | avoid sampling unfolding space |

**Appendix 1—table 3.** Average root-mean-square deviation (<$RMSD$> in units of ppm) between experimental CSs and those from the CS-restrained replica-averaged simulations.

| Nucleus | RUN3 | RUN4 | RUN5 | RUN6 | RUN7 | RUN8 | RUN9 |
|---|---|---|---|---|---|---|---|
| $C'$ | 0.833 | 0.655 | 0.776 | 0.854 | 0.793 | 0.907 | 0.727 |
| $C_\alpha$ | 1.055 | 0.879 | 0.929 | 1.065 | 0.940 | 1.103 | 0.894 |
| $N$ | 1.966 | 1.707 | 1.771 | 1.967 | 1.780 | 2.011 | 1.828 |
| $H_N$ | 0.379 | 0.275 | 0.291 | 0.368 | 0.284 | 0.414 | 0.286 |
| $H_A$ | 0.232 | 0.183 | 0.186 | 0.242 | 0.182 | 0.246 | 0.183 |

**Appendix 1—table 4.** Parameter set used in tunnel analysis using CAVER3.0 (*Chovancova et al., 2012*) and CAVER Analyst1.0 (*Kozlikova et al., 2014*).

| | |
|---|---|
| Minimum probe radius | 0.9 Å |
| Shell depth | 4 |
| Shell radius | 3 |
| Clustering threshold | 3.5 |
| Starting point optimization | |
| Maximum distance | 3 Å |
| Desired radius | 5 Å |

**Appendix 1—table 5.** Clustering analysis of tunnels (top three listed).

| Index | Population | Maximal bottleneck radius (Å) | Average bottleneck radius (Å) |
|---|---|---|---|
| #1 | 27% | 2.5 | 1.3 |
| #2 | 20% | 1.4 | 1.0 |
| #3 | 15% | 1.3 | 1.0 |

**Appendix 1—table 6.** Unbinding Pathways Explored by ABMD ($RMSD_{BNZ}$ as CV).

| | k=20 kJ/(mol · nm$^{-2}$) | | k=50 kJ/(mol · nm$^{-2}$) | |
|---|---|---|---|---|
| Index | Length | Path | Length | Path |
| RUN1 | 56 ns | P1 | 27 ns | P2 |
| RUN2 | 36 ns | P2 | 78 ns | P1 |

*Appendix 1—table 6 continued on next page*

*Appendix 1—table 6 continued*

|  | k=20 kJ/(mol · nm$^{-2}$) |  | k=50 kJ/(mol · nm$^{-2}$) |  |
| --- | --- | --- | --- | --- |
| RUN3 | 43 ns | P1 | 6 ns | P1 |
| RUN4 | 43 ns | P1 | 35 ns | P1 |
| RUN5 | 77 ns | P2 | 10 ns | P1 |
| RUN6 | 176 ns | P1 | 44 ns | P1 |
| RUN7 | 41 ns | P1 | 18 ns | P1 |
| RUN8 | 106 ns | P1 | 15 ns | P1 |
| RUN9 | 72 ns | P1 | 7 ns | P1 |
| RUN10 | 107 ns | P1 | 2 ns | P1 |
| RUN11 | 61 ns | P1 | 20 ns | P2 |
| RUN12 | 58 ns | P2 | 26 ns | P1 |
| RUN13 | 64 ns | P1 | 31 ns | P2 |
| RUN14 | 173 ns | P2 | 20 ns | P1 |
| RUN15 | 172 ns | P1 | 34 ns | P1 |
| RUN16 | 74 ns | P2 | 22 ns | P1 |
| RUN17 | 20 ns | P1 | 17 ns | P1 |
| RUN18 | 34 ns | P1 | 35 ns | P2 |
| RUN19 | 91 ns | P1 | 21 ns | P2 |
| RUN20 | 61 ns | P1 | 18 ns | P1 |
| Cost | 1.6 $\mu$s |  | 0.5 $\mu$s |  |
| Summary |  |  |  |  |
| P1 | 75% (15/20) |  | 75% (15/20) |  |
| P2 | 25% (5/20) |  | 25% (5/20) |  |

