## [Decision Letter]

Thank you for submitting your article "Mapping transiently formed and sparsely populated conformations on a complex energy landscape" for consideration by *eLife*. Your article has been reviewed by three peer reviewers, one of whom, Yibing Shan, is a member of our Board of Reviewing Editors, and another is James S Fraser (Reviewer #3), and the evaluation has been overseen by Michael Marletta as the Senior Editor. The reviewers have discussed the reviews with one another and the Reviewing Editor has drafted this decision to help you prepare a revised submission.

Summary:

In this manuscript, the authors employ state-of-art enhanced sampling techniques, including multiple variants of metadynamics, replica-averaged restrained MD and well-tempered replica-exchange MD, to conduct a systematic characterization of conformational dynamics of the L99A variant of T4L. This simulation study, which is shown to be in agreement with experimental measures of transition rates and kinetics, provides insights on the possible mechanism of conformational exchange between the ground state and the excited state. In particular, the excited state is shown to be conformationally broad and heterogeneous, and the transition path between the two states to be diverse. The work also identifies an key intermediate state for the transition, although its confirmation requires further experimental evidence.

Essential revisions:

Appreciating the thoroughness and the technical sophistication of the computation work and its robust connections with experimental data, the reviewers recognize the manuscript's value in benchmarking state-of-art simulation techniques and especially, in benchmarking conformational free energy calculations.

One of the more interesting parts of the of this study concerns the calculation of kinetic rate of the transition for L99A and the results is very encouraging. Because of its potential importance, this work would benefit greatly from additional calculations on mutants besides L99A. Specifically, the transition kinetics of L99A/G113A and L99A/G113A/R119P have been experimentally characterized. Calculating the kinetic rates of these two mutants and comparing the results with the experimental data would much strengthen case for the validity of such calculations.

The manuscript's attempt to elucidate the mechanism of ligand entrance is relatively weak. Further studies, for example, using simulations with ligand placed at the putative entrance to the channel in a partially open conformation would strengthen this part of the manuscript. If unsuccessful, the authors should consider weaken the related claims and de-emphasize the part of the work.

Relatedly, the reviewers think the manuscript's value lies more in benchmarking and validating state-of-art free energy calculation, than in providing significant new insight into the mechanism and function of T4L conformational transition. We would like to see revisions that de-emphasize the T4L specific claims and additional discussions with respect to benchmarking quantitative characterization of free energy landscapes using atomistic simulations.

---

## [Author Response]

*Essential revisions:*

*Appreciating the thoroughness and the technical sophistication of the computation work and its robust connections with experimental data, the reviewers recognize the manuscript's value in benchmarking state-of-art simulation techniques and especially, in benchmarking conformational free energy calculations.*

*One of the more interesting parts of the of this study concerns the calculation of kinetic rate of the transition for L99A and the results is very encouraging. Because of its potential importance, this work would benefit greatly from additional calculations on mutants besides L99A. Specifically, the transition kinetics of L99A/G113A and L99A/G113A/R119P have been experimentally characterized. Calculating the kinetic rates of these two mutants and comparing the results with the experimental data would much strengthen case for the validity of such calculations.*

We thank the reviewers for their thoughtful comments on our work. As discussed in more detail below, we agree that the benchmarking aspects of our work are important. We have therefore extended our work and performed additional simulations and analyses. In particular, we have calculated the kinetics of exchange of the L99A/G113A/R119P triple mutant. As in the case of the L99A variant, we found results that were in good agreement with experiment. Specifically, the rates of exchange are within a factor of 10 from the experimental rates, and the free energy difference estimated from the rates (-1.2 kcal/mol) is very similar to the value (-1.6 kcal/mol) obtained from the equilibrium sampling. These results are presented in the revised manuscript.

As suggested by the reviewers we have also performed extensive simulations of the L99A/G113A double mutant. At the end of a 1 microsecond long PathMetaD simulation, the free energy difference obtained is ~ -0.5 kcal/mol, in apparently good agreement with experiment (0.3 kcal/mol). In contrast to the results from L99A and the triple mutant we, however, find that this value appears not to be converged with substantial fluctuations and a (non-converged) free energy landscape that differs in shape from that of L99A and L99A/G113A/R119P. Thus, the calculated free energy difference is (in contrast to L99A and the triple mutant) more sensitive to how the major and minor states are separated. Thus, the good agreement that we observe could very well be fortuitous, and we have therefore opted not to include them in the revised manuscript. Instead we mention that we have performed these simulations, but that they did not converge.

*The manuscript's attempt to elucidate the mechanism of ligand entrance is relatively weak. Further studies, for example, using simulations with ligand placed at the putative entrance to the channel in a partially open conformation would strengthen this part of the manuscript. If unsuccessful, the authors should consider weaken the related claims and de-emphasize the part of the work.*

We agree that the link between the transiently opened tunnel and the pathway for ligand binding and release was weak (as indeed we stated). While we believe that a more quantitative study lies outside the scope of this work, we have now performed and include the results from a set of simulations aimed to demonstrate a more direct link between the tunnel observed in the apo protein and the pathway for ligand (un)binding. In particular, we have used so-called adiabatic biased molecular dynamics (ABMD) to study the mechanism by which the ligand escapes its internal binding site. In these simulations, a bias is introduced for the ligand to leave its binding site, but leaving the specific pathway to be determined by the molecular simulations. Of the 20 ligand unbinding events we observed in such simulations, 15 take a path located in the same region as the transiently opened tunnel we found in the apo protein. To examine the extent to which this result depends on the amount of bias used, we performed these simulations with two different strengths of the biasing force, and reassuringly found the same results in both cases (i.e. 15/20 in both cases). These new results are included in the manuscript. We now also explicitly show the tunnel induced by binding of an alkyl-benzene to the protein.

*Relatedly, the reviewers think the manuscript's value lies more in benchmarking and validating state-of-art free energy calculation, than in providing significant new insight into the mechanism and function of T4L conformational transition. We would like to see revisions that de-emphasize the T4L specific claims and additional discussions with respect to benchmarking quantitative characterization of free energy landscapes using atomistic simulations.*

We agree that a key result of our work lies in the testing and benchmarking of the ability of molecular simulations to map conformational free energy landscapes and determine the free energy and kinetics associated with conformational exchange. Prompted by the suggestion of the reviewers we have thus strengthened these aspects in several parts of the manuscript. This also includes the newly described kinetics for the triple mutant, as well as the fact that for both L99A and the triple mutant we find good agreement between the kinetic and equilibrium estimate of the free energy difference. In addition, we have toned down some of the more specific points that mostly relate to T4L. For example, we have now completely removed the section and results describing our studies of domain-domain motions in the minor state of L99A T4L. We have also shortened somewhat the description of the results relating to the mechanism of conformational exchange, and finally we have reorganized the sections to bring together the validation parts more coherently. We believe that these changes together make the focus of the paper stronger.